# V2P-Bench: Evaluating Video-Language Understanding with Visual Prompts for Better Human-Model Interaction

**Yiming Zhao**[1], **Yu Zeng**[1], **Yukun Qi**[1], **YaoYang Liu**[1], **Xikun Bao**[1],

**Lin Chen**[1], **Zehui Chen**[1], **Qing Miao**[3], **Chenxi Liu**[3], **Jie Zhao**[2], **Feng Zhao**[1]*

[1] University of Science and Technology of China
[2] Huawei Noah's Ark Lab
[3] Xi'an Jiaotong University

## Abstract

Large Vision-Language Models (LVLMs) have made significant strides in the field of video understanding in recent times. Nevertheless, existing video benchmarks predominantly rely on text prompts for evaluation, which often require complex referential language. To address this limitation, we propose **V2P-Bench**, a robust and comprehensive benchmark for evaluating the ability of LVLMs to understand **V**ideo **V**isual **P**rompts in human–model interaction scenarios. V2P-Bench consists of 980 videos and 1172 well-structured high-quality QA pairs, each paired with manually annotated visual prompt frames. The benchmark spans three main tasks and twelve categories, thereby enabling fine-grained, instance-level evaluation. Through an in-depth analysis of current LVLMs, we identify several key findings: *1)* Visual prompts are both more model-friendly and user-friendly in interactive scenarios than text prompts, leading to significantly improved model performance and enhanced user experience. *2)* Models are reasonably capable of zero-shot understanding of visual prompts, but struggle with spatiotemporal understanding. Even o1 achieves only 71.8%, far below the human expert score of 88.3%, while most open-source models perform below 60%. *3)* LVLMs exhibit pervasive hack phenomenon in video question answering, which intensify with longer videos and lower frame sampling density, artificially inflating performance scores. We anticipate that V2P-Bench will not only shed light on these challenges but also serve as a foundational tool for advancing human–model interaction. The code and datasets are available at `https://github.com/gaotiexinqu/v2p-bench`.

## 1 Introduction

Large Vision-Language Models (VLMs) have recently achieved remarkable success across a wide range of multimodal tasks, including image captioning (Zeng et al., 2025b), visual question answering (Qi et al., 2025; Pan et al., 2025; Huang et al., 2025; 2026a), self-supervision (Zeng et al., 2025a), deep research (Huang et al., 2026b; Zeng et al., 2026), document understanding (Wang et al., 2025), multimodal generation (Zhang et al., 2025b;c; Ma et al., 2025; Zhang et al., 2025a), retrieval-augmented generation (Wang et al., 2026), etc. further extends the model's versatility across general tasks. Notable models like Gemini-2.5-Pro (Team et al., 2024) and LLaVA-Video (Zhang et al., 2024b) have set new performance benchmarks. In response, numerous benchmarks have emerged to evaluate these models comprehensively across diverse tasks (Li et al., 2024c; Mangalam et al., 2023; Fu et al., 2024). These benchmarks support LVLM evaluation, enabling nuanced assessments of their strengths and weaknesses in real-world applications. This expanding landscape fosters rigorous testing and drives further innovation in LVLM development.

However, most benchmarks utilize texts for human-model interaction, which inevitably introduces certain inherent limitations. As shown in Figure 1, text prompts usually fail to provide precise spatial

---

* Corresponding author

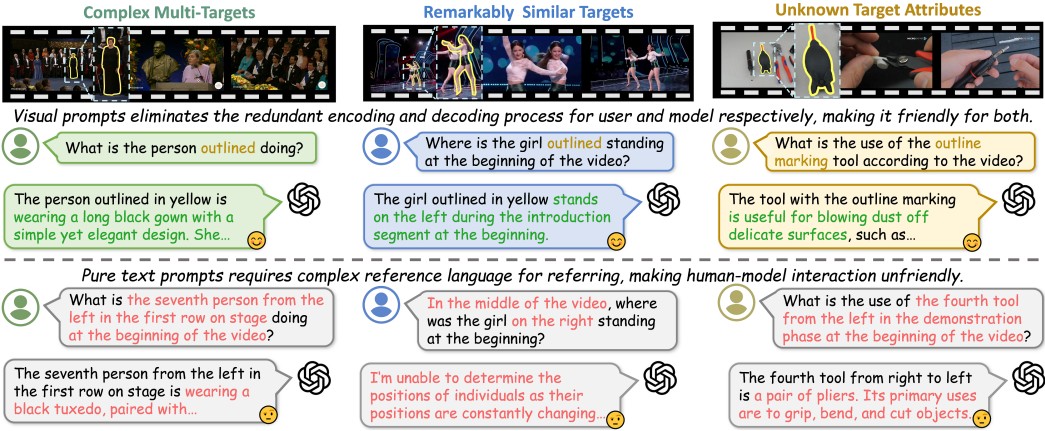

Figure 1: **Comparison of text prompts and visual prompts for users and models.** Pure text prompts suffer from complex encoding and decoding, requiring users to describe targets with intricate language and models to infer the intended referents, which often leads to misalignment, especially in complex scenes with multiple or similar targets. In contrast, visual prompts directly indicate targets on video frames, providing a more intuitive and user- and model-friendly interaction.

and temporal references, resulting in difficulties when assessing the ability of LVLMs to understand specific areas or moments in videos, particularly in complex multi-object scenarios. For users, a significant amount of referential language is required to specify targets. For the model, it first needs to comprehend the user's referential language, making it prone to confusion at the initial step.

In contrast, as a frontier approach to multimodal human-model interaction, visual prompts offer a simpler and more precise mechanism that enhances model understanding and better aligns with human intuitive cognition. Prior works (Cai et al., 2024; Yang et al., 2023) explore visual prompts in the image domain and demonstrate their advantages over text-based prompts. However, limited studies in the video modality hinder further advances in multimodal human-model interaction.

To bridge this gap, we propose **V2P-Bench**, a comprehensive benchmark for evaluating LVLMs' video understanding in human-model interaction scenarios. As shown in Figure 2, V2P-Bench covers three main tasks, twelve categories, twenty video types, eight visual prompt types, and durations ranging from three seconds to two hours. Each query contains a single visual prompt annotation, emphasizing fine-grained spatial and temporal understanding to comprehensively assess LVLMs' video capabilities. All videos are manually curated to ensure high quality.

We conduct a comprehensive evaluation on V2P-Bench. A user study comparing visual and text prompts shows that visual prompts significantly improve both model performance and human–model interaction. We further evaluate 15 LVLMs, including 3 closed-source and 12 open-source models. Results reveal that even state-of-the-art models perform poorly (e.g., o1 (OpenAI, 2024) achieves 71.8%), far below the human expert score of 88.3%, highlighting limitations in video visual prompt understanding. Additional analysis exposes widespread hacking behaviors in video question answering, which worsen with longer videos and lower frame sampling rates.

In a nutshell, our contributions are as follows:

- V2P-Bench has been meticulously designed, comprising twelve categories covering a wide range of video types and diverse visual prompts. Collection and annotation process undergoes rigorous human validation, aiming to provide the community with a high-quality benchmark for multi-model human-model interaction.

- We demonstrate the superiority of visual prompts over text prompts. Experimental results reveal that current models exhibit substantial limitations in comprehending video visual prompts and display evidence of hacking behaviors in video question-answering tasks.

- V2P-Bench pioneeringly applies visual prompts in video understanding evaluation for human-model interaction, addressing critical limitations in existing text-based evaluation frameworks. We seek to advance the field of video visual prompt understanding evaluation and establish a foundation for more intuitive human-model interaction.

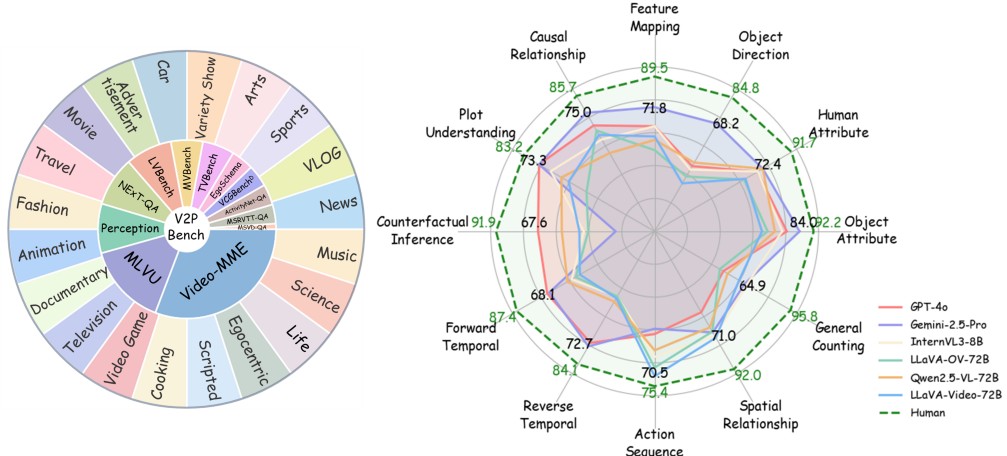

Figure 2: *(Left)* **Datasets and categories.** Our dataset is derived from twelve datasets and contains twenty restructured categories. *(Right)* **Performance radar chart.** We report the performance of different models on V2P-Bench by dimension. SOTA for each dimension is given.

## 2 RELATED WORK

### 2.1 LVLMs FOR VIDEO UNDERSTANDING

The rapid development of image-based LVLMs (Liu et al., 2024b; 2023; 2024a; Li et al., 2024a; Chen et al., 2024a;d; Bai et al., 2023) has significantly enhanced the potential of video understanding and question answering tasks, injecting new vitality into the field of artificial intelligence. VideoChat (Li et al., 2023b) and Video-ChatGPT (Maaz et al., 2023) are preliminary attempts in the realm of video understanding. Notable recent works include CogVLM2-Video (Hong et al., 2024), InternVL2 (Chen et al., 2024d) and LLaVA-Video (Zhang et al., 2024b), which treat videos as sequences of images and leverage the powerful image comprehension capabilities to process video modality. Furthermore, the high computational and memory demands required for handling high frame rates and long videos have spurred advancements in video compression technologies. For instance, InternVideo2 (Wang et al., 2024c) and Video-LLaMA (Zhang et al., 2023) utilize QFormer (Li et al., 2023a) for efficient video feature extraction. Despite promising results, current LVLMs primarily rely on text prompts and still face challenges in fine-grained spatial and temporal understanding when given visual prompts as input.

### 2.2 VIDEO UNDERSTANDING BENCHMARKS

Traditional video understanding benchmarks, such as MSRVTT-QA (Xu et al., 2017), ActivityNet-QA (Yu et al., 2019), and NExT-QA (Xiao et al., 2021), focus on basic action recognition and video question answering. Recently, more benchmarks have been proposed. MMBench (Liu et al., 2024c), SEED-Bench (Li et al., 2024b), and MVBench (Li et al., 2024c) mainly concentrate on short video clips for evaluation. EgoSchema (Mangalam et al., 2023) and MovieQA (Tapaswi et al., 2016) provide insights into narrative and thematic understanding. LongVideoBench (Wu et al., 2024), Video-MME (Fu et al., 2024), and LVBench (Wang et al., 2024b) offer longer videos and a broader variety of tasks. Additionally, recent works like INST-IT (Peng et al., 2024) and VideoRefer (Yuan et al., 2024) have introduced instance-level video question answering benchmarks. However, constrained by insufficiently robust and comprehensive, they still fail to adequately simulate real-world interactions. To address this limitation, we introduce V2P-Bench, allowing for a comprehensive evaluation of LVLMs that simulates multimodal human-model interaction in realistic settings.

### 2.3 VISUAL PROMPT AS A USER-FRIENDLY SOLUTION

Compared to text prompts, visual prompts offer a simple and effective means of facilitating interaction between users and models. Visual prompts have been widely utilized in image understanding. ViP-LLaVA (Cai et al., 2024) enhances the ability of LVLMs to comprehend local image regions by overlaying arbitrary visual prompts on images. Draw-and-Understand (Lin et al., 2024) employs a

Table 1: **Comparison of different datasets. Answer Type** indicates whether the QA pair is open-ended(OE) or multiple-choice(MC). **Multi Level** represents whether the videos cover multiple duration levels. **Open Domain** indicates whether the video source is diversified. **Visual Prompt** represents whether the video contains visual prompts. Refer to Appendix A for more features.

| Benchmarks | Videos | Samples | Tasks | Avg duration | Annotation | Answer Type | Multi Level | Open Domain | Visual Prompt |
|---|---|---|---|---|---|---|---|---|---|
| MSVD-QA(Xu et al., 2017) | 504 | 13157 | 1 | 9.8s | Auto | OE | ✗ | ✗ | ✗ |
| MSRVTT-QA(Xu et al., 2017) | 2990 | 72821 | 1 | 15.2s | Auto | OE | ✗ | ✗ | ✗ |
| ActivityNet-QA(Yu et al., 2019) | 800 | 8000 | 3 | 111.4s | Manual | OE | ✗ | ✗ | ✗ |
| NExT-QA(Xiao et al., 2021) | 1000 | 8564 | 3 | 39.5s | Manual | MC | ✗ | ✗ | ✗ |
| Perception Test(Patraucean et al., 2024) | 11600 | 44000 | 4 | 23.0s | Auto&Manual | MC | ✗ | ✗ | ✗ |
| MLVU(Zhou et al., 2024) | 1334 | 2593 | 9 | ~12min | Auto&Manual | OE&MC | ✓ | ✓ | ✗ |
| VCGBench-Diverse(Maaz et al., 2024) | 877 | 4354 | 6 | 217.0s | Auto&Manual | OE | ✗ | ✓ | ✗ |
| MVBench(Li et al., 2024c) | 3641 | 4000 | 20 | 16.0s | Auto | MC | ✗ | ✓ | ✗ |
| HourVideo(Chandrasegaran et al., 2024) | 500 | 12976 | 18 | 45.7min | Auto&Manual | MC | ✗ | ✗ | ✗ |
| LVBench(Wang et al., 2024b) | 103 | 1549 | 6 | 68.4min | Manual | MC | ✗ | ✓ | ✗ |
| EgoSchema(Mangalam et al., 2023) | 5063 | 5063 | 1 | 180.0s | Auto | MC | ✗ | ✗ | ✗ |
| Video-MME(Fu et al., 2024) | 900 | 2700 | 12 | 17.0min | Manual | MC | ✓ | ✓ | ✗ |
| INST-IT Bench(Peng et al., 2024) | 206 | 1000 | 1 | 14.2s | Auto&Manual | OE&MC | ✗ | ✗ | ✓ |
| VideoRefer Bench[Q](Yuan et al., 2024) | 198 | 1000 | 5 | 13.8s | Manual | MC | ✗ | ✗ | ✓ |
| **V2P-Bench(ours)** | 980 | 1172 | 12 | 19.0min | Manual | MC | ✓ | ✓ | ✓ |

two-stage training approach to improve performance in pixel-level tasks. Set-of-Mark (Yang et al., 2023) introduces a novel visual prompting method to enhance the performance of LVLMs in visual localization tasks. However, research on visual prompts in the context of video remains limited. INST-IT (Peng et al., 2024) introduces instruction tuning with visual prompts to enhance instance-level understanding in LVLMs. VideoRefer Suite (Yuan et al., 2024) creates a large instance-level video instruction dataset to assist LVLMs in understanding spatiotemporal information in videos.

## 3 V2P BENCH

Table 1 compares the key difference of V2P-Bench with previous benchmarks. The first two blocks list traditional pure text video understanding benchmarks, which primarily understand videos at a holistic level and lack instance-level comprehension. Instance-level understanding is crucial as it focuses on the specific elements of greatest interest to us, requiring a more nuanced understanding and consistency.

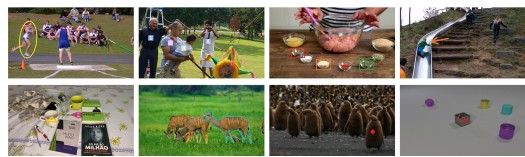

Figure 3: Various visual prompt types.

As shown in the third block, although INST-IT Bench (Peng et al., 2024) and VideoRefer Bench[Q] (Yuan et al., 2024) use visual prompts for question-answering, their: *1)* visual prompts are annotated on all frames, rendering them unsuitable for human–model interaction scenarios; *2)* all data are sourced exclusively from VIS datasets (Yang et al., 2019; Pont-Tuset et al., 2017; Ding et al., 2023), thereby exhibiting limitations in both robustness and comprehensiveness, meaning *a)* Shorter video durations( 14.2s and 13.8s); *b)* Single continuous shots; *c)* Limited video sources; *d)* Objects of interest not be suitable for question-answering.

### 3.1 TASK DEFINITION

To facilitate fine-grained evaluation of LVLMs from various perspectives, we categorize the questions according to dimensions. Our dimension design strives to ensure both comprehensiveness and orthogonality, and ultimately includes three main tasks and twelve dimensions. Definitions for tasks and dimensions are as follows:

• **Basic Perception** focuses on understanding the intrinsic attributes of objects and humans in the visual prompt. This task includes: *1)* Object Attribute (OA); *2)* Human Attribute (HA).

• **Temporal Understanding** emphasizes comprehension and processing of dynamic information and chronological sequences in videos. This task includes: *1)* Object Direction (OD); *2)* Feature Mapping (FM); *3)* Forward Temporal (FT); *4)* Reverse Temporal (RT); *5)* Action Sequence (AS); *6)* Spatial Relationship (SR); *7)* General Counting (GC).

• **High-level Reasoning** extends beyond perception and temporal understanding, requiring logical inference and judgment to derive new conclusions or answers. This task includes: *1)* Causal Relationship (CR); *2)* Plot Understanding (PU); *3)* Counterfactual Inference (CI).

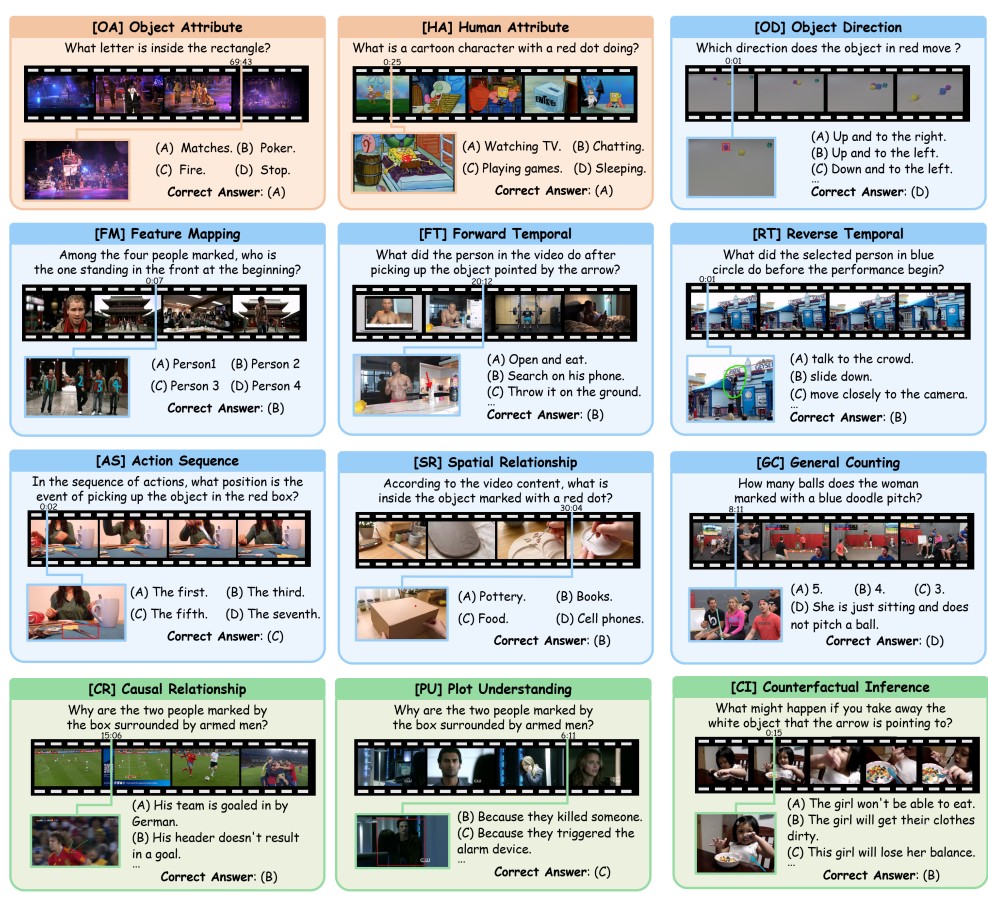

Figure 4: Examples for each dimension of V2P-Bench.

Detailed elaborations and examples of each dimension are provided in Appendix A.2.

## 3.2 DATASET CONSTRUCTION

### 3.2.1 VIDEO COLLECTION

To create our dataset, we start from existing video benchmarks, as they already have a wide distribution of durations and diverse video types. We categorize the video durations into short, medium, and long videos. Additionally, we reclassify all the videos, resulting in twenty video categories, as shown in Figure 2*(left)*. Our final dataset covers multiple video domains while maintaining a relative balance in video lengths.

### 3.2.2 QA AND VISUAL PROMPT ANNOTATION

After completing the collection process, we conduct the annotation of QA pairs and visual prompts to evaluate the capabilities of LVLMs in video understanding with visual prompts. The annotation work is carried out by researchers proficient in English. To ensure data quality, we provide thorough training for the annotators and conduct pilot annotations to assess their annotation capabilities.

While annotating the QA pairs, annotators are also required to perform visual prompt annotations. To better approximate real-world distributions, we adopt a fully manual approach for annotating visual prompts, with each QA pair constrained to one visual prompt frame. We predefine various types of visual prompts as follows: rectangle, mask contour, ellipse, triangle, scribble, point, arrow and SoM, as shown in Figure 3.

### 3.2.3 POST PROCESSING

To ensure the quality of the dataset, we conduct a rigorous review of the annotated data after completion, including both VLMs and manual review processes.

**Blind LLMs Filtering.** Inspired by MMStar (Chen et al., 2024b), we perform plain text filtering on the QA pairs to ensure that questions could only be answered correctly by viewing the videos. Specifically, we provide only the pure text QA pairs to the most powerful closed-source models GPT-4o and Gemini-2.5-Pro. We set the sampling temperature to 0.2 and conduct two rounds of inference, then exclude samples for which both rounds provided correct answers.

**Manual and Rule-based Review.** After that, we perform a rule-based check and manual review of the remaining data. We exclude samples where the length disparity between different options was too significant. Additionally, we shuffle the order of multiple-choice options to ensure a uniform distribution of answer choices, thereby eliminating potential biases of different models toward specific options. The filtering ratios are reported in the Appendix A.4. The final balanced proportions of the four options are 28.0%, 23.9%, 25.0% and 23.1%. Through this rigorous dataset construction process, we strive to provide a high-quality, diverse, and balanced dataset that will benefit researchers in the field of human-model interaction.

## 4 EXPERIMENTS

### 4.1 EXPERIMENT SETUP

**Evaluation Models.** To thoroughly evaluate the effectiveness of V2P-Bench, we conduct assessments on multiple models, including 3 closed-source models: o1 (OpenAI, 2024), GPT-4o (Hurst et al., 2024) and Gemini-2.5-Pro (Team et al., 2024); 12 open-source models: LLaVA-OneVision(7B 72B) (Li et al., 2024a), InternVL3-8B (Zhu et al., 2025), mPLUG-Owl3-7B (Ye et al., 2024), LLaVA-Video(7B 72B) (Zhang et al., 2024b), MiniCPM-V 2.6-8B (Yao et al., 2024), Qwen2.5-VL(7B 72B) (Wang et al., 2024a), MiMo-VL-7B (Team et al., 2025), LLaVA-NeXT-7B (Liu et al., 2024a) and LLaVA-NeXT-INST-IT-7B (Peng et al., 2024). This essentially covers the mainstream LVLMs currently available.

**Implementation Details.** For open-source models, we select the sampling frame rate based on the context length of each model. Specifically, we average 128 frames from the video for LLaVA-OneVision(7B 72B), InternVL3-8B, mPLUG-Owl3-7B, LLaVA-Video(7B 72B), MiniCPM-V 2.6-8B, Qwen2.5-VL(7B 72B), MiMo-VL-7B, 4 frames for LLaVA-NeXT-7B and LLaVA-NeXT-INST-IT-7B. For o1 and GPT-4o, we average 64 frames. For Gemini-2.5-Pro, the raw video was uploaded directly without audio track. The visual prompt frame is placed after the video for all models. For other hyperparameters, we follow the settings in VLMEvalKit (Duan et al., 2024).

### 4.2 QUANTITATIVE RESULTS

Table 2 and 5 present the comprehensive evaluation results of V2P-Bench across different dimensions and video durations. These results encompass the performance of human experts, the blind answering task, and evaluations of 15 different models. We can conclude that o1 achieves the highest overall score; however, its performance is not consistently superior across all dimensions, particularly in Object Direction and Action Sequence. As shown at the top of Table 2, human experts achieve an accuracy of 88.3%, representing the upper bound of human performance. For the blind answering task, we observe that all three models perform below 10% on this task, with GPT-4o at 1.4%, Gemini-2.5-Pro at 9.6%, and Qwen2.5-VL-7B at 3.0%. This demonstrates that our post-processing pipeline effectively filters out commonsense-based questions, thereby ensuring high quality and robustness.

Below we summarize our key findings as follows:

***Which prompt type works better for humans and models?*** Most benchmarks rely solely on text prompts, requiring models to answer questions based on textual questions and visual context. To investigate which type of prompt(textual or visual) is more conducive in human model interaction scenarios, we conduct a prompt comparison experiment and a real-user study. Results are reported in

Table 2: **Evaluation Results on V2P-Bench across Dimensions.** We report results for 12 open-source models, 3 closed-source models, 3 blind answering results and human performance on V2P-Bench across dimensions. The best results are **bold** and the second-best are underlined.

| Method | OA | HA | OD | FM | CR | PU | CI | FT | RT | AS | SR | GC | Avg |
|---|---|---|---|---|---|---|---|---|---|---|---|---|---|
| Human Performance | 92.2 | 91.7 | 84.8 | 89.5 | 85.7 | 83.2 | 91.9 | 87.4 | 84.1 | 75.4 | 92.0 | 95.8 | 88.3 |
| *Pure Text as Input* | | | | | | | | | | | | | |
| GPT-4o | 2.3 | 1.2 | 0.0 | 0.0 | 4.6 | 1.8 | 0.0 | 1.7 | 0.0 | 0.0 | 0.0 | 4.6 | 1.4 |
| Gemini-2.5-Pro | 15.6 | 12.0 | 6.5 | 3.9 | 9.2 | 10.9 | 8.1 | 5.2 | 13.6 | 1.8 | 8.0 | 7.4 | 9.6 |
| Qwen2.5-VL-72B | 3.9 | 4.0 | 0.0 | 0.0 | 7.3 | 3.6 | 0.0 | 0.0 | 6.8 | 5.4 | 0.0 | 0.0 | 3.0 |
| *Closed-source Models* | | | | | | | | | | | | | |
| o1 | **85.2** | **78.4** | 23.1 | **78.1** | 71.6 | **78.7** | 66.7 | **69.1** | **73.1** | 50.0 | 64.1 | 51.2 | **71.8** |
| GPT-4o | 76.6 | 68.9 | 41.3 | 60.8 | 67.0 | 73.3 | 67.6 | 68.1 | 70.5 | 50.0 | 54.0 | 48.4 | 65.4 |
| Gemini-2.5-Pro | 84.0 | 72.4 | **68.2** | 71.8 | 75.0 | 73.3 | 22.6 | 66.7 | 72.7 | 47.4 | 67.5 | 63.6 | 69.8 |
| *Open-source Models* | | | | | | | | | | | | | |
| LLaVA-NeXT-7B | 56.6 | 55.6 | 34.8 | 52.5 | 43.0 | 48.6 | 31.6 | 42.6 | 42.2 | 28.1 | 42.0 | 30.5 | 46.0 |
| LLaVA-NeXT-INST-IT-7B | 57.4 | 58.4 | 26.1 | 42.4 | 43.0 | 49.2 | 31.6 | 49.2 | 42.2 | 26.3 | 42.0 | 27.4 | 46.3 |
| LLaVA-OV-7B | 57.1 | 52.1 | 28.3 | 47.1 | 63.8 | 59.1 | 41.0 | 42.1 | 35.6 | 63.2 | 62.8 | 43.2 | 52.8 |
| LLaVA-OV-72B | 65.5 | 59.9 | 34.7 | 47.0 | 63.8 | 43.2 | 38.5 | 50.0 | 41.1 | 66.3 | 66.9 | 45.9 | 56.7 |
| InternVL3-8B | 73.9 | 69.1 | 39.1 | 60.8 | 58.1 | 65.9 | 41.0 | 52.6 | 41.1 | 61.1 | 69.7 | 64.9 | 61.7 |
| mPLUG-Owl3-7B | 61.3 | 54.4 | 28.3 | 49.0 | 60.0 | 50.0 | 51.3 | 60.5 | 34.4 | 55.8 | 58.6 | 37.8 | 52.6 |
| LLaVA-Video-7B | 60.5 | 58.1 | 37.0 | 49.0 | 62.9 | 54.5 | 41.0 | 52.6 | 48.9 | 57.9 | 56.6 | 40.5 | 54.8 |
| LLaVA-Video-72B | 62.2 | 60.8 | 30.4 | 54.9 | 61.0 | 54.5 | 43.6 | 47.4 | 42.2 | **70.5** | **71.0** | 59.5 | 58.6 |
| MiniCPM-V 2.6-8B | 68.9 | 59.4 | 26.1 | 56.9 | 58.1 | 50.0 | 33.3 | 50.0 | 34.4 | 57.9 | 67.6 | 43.2 | 55.3 |
| Qwen2.5-VL-7B | 60.5 | 56.7 | 17.4 | 45.1 | 47.6 | 40.9 | 48.7 | 52.6 | 32.2 | 47.4 | 50.3 | 35.1 | 48.1 |
| Qwen2.5-VL-72B | 69.7 | 72.4 | 43.5 | 52.9 | 49.5 | 59.1 | 53.8 | 55.3 | 44.4 | 57.9 | 64.1 | 51.4 | 59.8 |
| MiMo-VL-7B | 67.2 | 57.6 | 37.0 | 45.1 | 47.6 | 52.3 | 59.0 | 55.3 | 41.1 | 50.5 | 61.4 | 43.2 | 53.8 |

Table 3. We can draw the following observations: **(1) Visual prompts are more model-friendly than text prompts.** As shown in Table 3a, simply converting visual prompts into text prompts leads to a substantial drop in accuracy across all participating models, with the most pronounced decline 15.1% observed in Gemini-2.5-Pro. This is because text prompts require the model to decode the target from the text, which increases the difficulty of comprehension. Moreover, textual references can sometimes be ambiguous, as illustrated in Figure 1. In contrast, visual prompts can precisely localize the target within video frames, bypassing both the user's need to encode intentions in text and the model's need to decode them. **(2) Visual prompts are more user-friendly than text prompts.** We recruit 20 volunteers to participate in the experiment. Specifically, the interaction process consists of: watching a video, formulating a meta-question, rewriting the question into both text and a visual-prompt version, completing the QA session, and then indicating their preference(text or visual prompts). We also record the order in which users write the text version and the visual prompt version of each question. Each participant is required to produce 10 text and 10 visual prompt questions. All questions are open-ended but designed to have unique correct answers to minimize potential subjectivity or evaluation bias. The correctness of model responses is assessed by our annotators. All videos are randomly sampled from the V2P-Bench dataset we construct. Throughout the study, Gemini-2.5-Pro serves as the conversational agent. To systematically evaluate performance, we record five key metrics: answer accuracy (max 100), completion time (seconds), user satisfaction (max 10), user preference, and question order. As shown in Table 3b, users complete tasks more quickly in the visual prompt interaction setting, with the time reduced from 25.2s to 18.1s, saving 7.7s on average. In addition, the improvement in model performance means more satisfactory responses for users. Together, these factors contributed to an average user satisfaction score of 8.2, 2.4 points higher than under the text-only prompt condition. This indicates that visual prompts provide clear

Table 3: Comparison of text and visual prompts for humans and models.

(a) Model performance.

| Model | Text | Visual |
|---|---|---|
| GPT-4o | 53.0 | **65.4** |
| Gemini-2.5-Pro | 54.7 | **69.8** |
| LLaVA-Video-7B | 42.4 | **54.8** |
| Qwen2.5-VL-7B | 43.1 | **52.4** |
| Mimo-VL-7B | 46.7 | **55.6** |

(b) User experience study.

| Metric | Text | Visual |
|---|---|---|
| Acc | 57.0 | **69.5** |
| Cost Time | 25.2 | **18.1** |
| User Satisfaction | 5.3 | **7.5** |

(c) User preference for prompt type.

| Preference | Text | Visual | None |
|---|---|---|---|
| Nums | 57 | **129** | 14 |
| Percentage | 28.5% | **64.5%** | 7.0% |

(d) Question order on user responses.

| Order | Text First | Visual First |
|---|---|---|
| Nums | 64 | **136** |
| Percentage | 34.0% | **68.0%** |

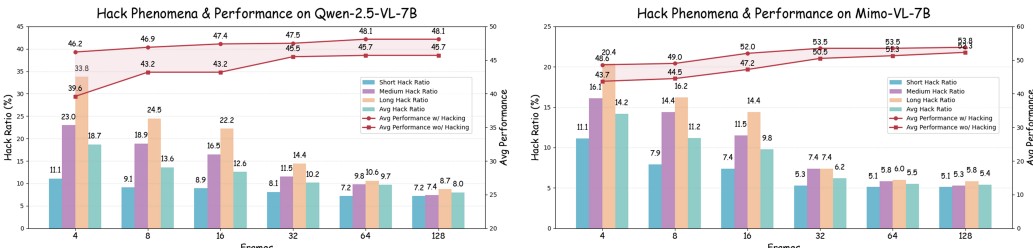

Figure 5: Hack Phenomena and Model Performance on V2P-Bench.

advantages in human–model interaction, improving both overall satisfaction and efficiency. Besides, 64.5% of participants explicitly preferred using visual prompts, while only 28.5% preferred text-only input, and 7.0% had no preference. What's more, 68% of the questions were initiated using visual prompts. This indicates that for real users, visually selecting the target is more intuitive and less effortful than crafting a natural-language description, especially for objects, locations, or people that are difficult to describe precisely with text. Refer to Appendix E.2 for the implementation details.

***Can the models effectively comprehend visual prompts, and how do they perform?*** In Table 2, we report the performance of human experts and all models across dimensions, leading to the following observations: **(1) Models are reasonably capable of zero-shot understanding of visual prompts.** Except for LLaVA-NeXT-INST-IT-7B, the open-source models have not been trained on visual prompt datasets, yet all achieve performance above 45 points, surpassing the random-guess baseline (25%) by over 20%. Notably, in the Basic Perception task, every model achieves over 50%. This phenomenon arises because LVLMs are typically trained on massive image–text and video–text pairs, enabling them to learn broad visual–semantic associations and thus exhibit strong zero-shot transfer capabilities. **(2) Models struggle with spatiotemporal understanding.** Object Direction requires models to identify the direction of an object's movement, while Spatial Relationship requires models to understand the dynamic spatial positions. The average scores on the two dimensions are only 34.4% and 45.7%, respectively, indicating that spatiotemporal tasks remains a weakness of current models and requires further improvement. **(3) Closed-source models and large-scale parameter models possess stronger capabilities.** As shown in Table 2, three closed-source models outperform all open-source counterparts. This disparity highlights the persistent challenges in advancing open-source models. Moreover, for the same open-source model with varying scales, for example Qwen2.5-VL 7B and 72B, the larger-parameter variant demonstrates stronger capabilities and achieves higher scores, which is consistent with the established scaling laws of LVLMs.

***Does the expert model exhibit the anticipated superior performance?*** LLaVA-NeXT-INST-IT is fine-tuned on INST-IT dataset based on LLaVA-NeXT, yet the results show only a marginal improvement of 0.3%. There are two main reasons for this limited performance gain: **(1) Limited diversity of visual prompt types.** LLaVA-NeXT-INST-IT is trained exclusively on SoM data. As observed in Table 13, while the model achieves a 15.0% improvement on SoM prompts, its performance drops on other types of visual prompts, indicating a forgetting phenomenon. This suggests that models' training should cover a broad range of visual prompt types to better meet the demands of real-world scenarios. **(2) Differences in data format.** Unlike V2P-Bench, INST-IT dataset annotates visual prompts on every sampled frame, as shown in Figure 9 *(left)*. This redundant annotation does not account for the constraints of user interactions in real human–model interaction scenarios, which contributes to the suboptimal performance. A detailed analysis of this phenomenon provides important insights for building future datasets and training strategies. Refer to Appendix D for more information.

***Are vision-language models truly understanding videos, or merely exploiting hacks?***

Due to the sparsity of frame sampling and the upper limits of model perception, the models may guess answers rather than rely on visual context. We randomly shuffle the videos and questions and performed inference with Qwen2.5-VL-7B and MiMo-VL-7B. As shown in Table 4, only 6.4% and 3.9% of the cases trigger a refusal, respectively, while all others follow the instructions to select an option. This indicates that current models are trained to be "test-takers" often neglecting basic factual information. To investigate to what extent models exhibit hack behavior

Table 4: Results with shuffled video and question pairs.

| Model | Trigger Ratio |
|---|---|
| Qwen2.5-VL-7B | 6.4% |
| MiMo-VL-7B | 3.9% |

on V2P-Bench, we conduct experiments in which the models are instructed to reject when they cannot reach a conclusion. Results are reported in Figure 5. From our analysis, we draw the following conclusions: **(1) Hack Phenomena exist in video benchmark evaluations.** Both Qwen2.5-VL-7B and MiMo-VL-7B exhibit positive Hack Ratios across all settings, accompanied by varying degrees of performance degradation. **(2) Longer videos exacerbate Hack Phenomena.** For instance, under a 4-frame sampling setting, the Hack Ratios of Qwen2.5-VL-7B for short, medium, and long videos are 11.1%, 23.0%, and 33.8%, respectively. This pattern is consistent across other experiments as well. **(3) Fewer sampled frames increase Hack Phenomena.** For example, when reducing the number of sampled frames for Qwen2.5-VL-7B from 128 to 4, the average Hack Ratio steadily rises from 8.0% to 18.7%. The presence of Hack Phenomena can be attributed to three factors: **(1) Insufficient information**, as existing sparse sampling strategies fail to provide the model with enough information to get the answer; **(2) Limited model perception**, where excessive visual context may overwhelm the key information; **(3) Training strategy.** Models are trained as instruction-following agents, leading them to prioritize following instructions over grounding their responses in factual information. Efforts should be made to actively enhance model capabilities and explore novel visual representation mechanisms that capture a broader range of effective context. When visual context is insufficient, models should proactively request clarification. We also report the results of other benchmarks in Appendix I.3, which are consistent with our observation on V2P-Bench.

***How LVLMs are robust to varied video duration?***

In Table 5, we compare the performance of different models on short, medium, and long videos, from which we can conclude that: **Performance degrades as video length increases.** All models exhibit a significant drop in performance. For example, o1's performance on long videos decreases by 23.5% compared to the medium. This decline is primarily due to the increased sparsity of frame sampling as video length grows, which reduces the amount of effective visual content. Such sparsity prevents models from retaining all relevant visual–semantic information, thereby hindering accurate predictions. However, all models perform worse on short videos compared to medium-length videos unexpectedly. This is likely because over half of the short videos are drawn from Perception Test, MVBench and TVBench, which have high information density and contain challenging questions related to spatiotemporal questions.

Table 5: Evaluation results on V2P-Bench across durations. The best results are **bold** and the second-best are underlined.

| Method | Short | Medium | Long | Avg |
|---|---|---|---|---|
| Human Performance | 91.6 | 87.3 | 84.0 | 88.3 |
| *Pure Text as Input* | | | | |
| GPT-4o | 1.1 | 1.9 | 1.6 | 1.4 |
| Gemini-2.5-Pro | 10.0 | 10.6 | 8.1 | 9.6 |
| Qwen2.5-VL-72B | 2.7 | 5.1 | 2.5 | 3.0 |
| *Closed-source Models* | | | | |
| o1 | **75.2** | 83.9 | **60.4** | **71.8** |
| GPT-4o | 67.3 | 70.8 | 59.3 | 65.4 |
| Gemini-2.5-Pro | 73.8 | **86.3** | 54.5 | 69.8 |
| *Open-source Models* | | | | |
| LLaVA-NeXT-7B | 47.0 | 47.1 | 43.8 | 46.0 |
| LLaVA-NeXT-INST-IT-7B | 48.6 | 51.1 | 39.5 | 46.3 |
| LLaVA-OV-7B | 51.3 | 63.0 | 47.3 | 52.8 |
| LLaVA-OV-72B | 54.5 | 70.4 | 49.0 | 56.7 |
| InternVL3-8B | 61.7 | 68.5 | 55.6 | 61.7 |
| mPLUG-Owl3-7B | 52.2 | 62.5 | 44.9 | 52.6 |
| LLaVA-Video-7B | 54.1 | 65.7 | 46.5 | 54.8 |
| LLaVA-Video-72B | 57.5 | 66.2 | 54.3 | 58.6 |
| MiniCPM-V 2.6-8B | 53.3 | 66.2 | 50.2 | 55.3 |
| Qwen2.5-VL-7B | 48.0 | 53.2 | 43.6 | 48.1 |
| Qwen2.5-VL-72B | 62.4 | 63.9 | 50.2 | 59.8 |
| MiMo-VL-7B | 56.8 | 58.8 | 42.4 | 53.8 |

***How does the structure of visual prompts affect model performance?*** To investigate the effect of the visual query structure itself on the performance of visual prompts, we randomly sampled 217 data instances. For each question–answer pair, we annotated the corresponding visual prompt frames with multiple types of visual prompts, while strictly keeping all other settings identical, including the number of frames, prompts. The results are shown in Table 6.

For the same visual prompt type, the doodle-style shapes are slightly weaker than the standard shapes. When switching from standard shapes to doodle shapes, the performance of Qwen2.5-VL-7B and MiMo-VL-7B

Table 6: Performance on Different Visual Prompts.

| Method | Standard Shapes | | | Doodle Shapes | | |
|---|---|---|---|---|---|---|
| | Rectangle | Arrow | SoM | Rectangle | Arrow | SoM |
| Qwen2.5-VL-7B | 47.3 | 43.6 | 45.1 | 46.7 | 42.9 | 44.4 |
| MiMo-VL-7B | 54.2 | 51.2 | 52.7 | 53.6 | 50.3 | 51.9 |

decreases by 0.7% and 0.8%, respectively. This is reasonable, since most training data are synthetic, whereas hand-drawn doodles often have unstable boundaries, making it harder for the model to extract consistent visual prompts. Regardless of whether standard or doodle shapes are used, the overall performance trend remains Rectangle > SoM > Arrow. Among these, rectangles appear most frequently in training data and can fully enclose the target region, providing a stable and explicit spatial localization signal. In contrast, arrows are typically smaller and carry lower information density, making it more difficult for the model to capture key spatial relationships and therefore resulting in the weakest performance.

## 4.3 ERROR ANALYSIS

In this section, we analyze the error patterns of LVLMs in video visual prompt understanding. We examine and categorize 470 model predictions from Qwen2.5-VL-7B, identifying three main representative error types. These patterns are illustrated in Figure 10 to Figure 12, and their distribution is presented in Figure 6.

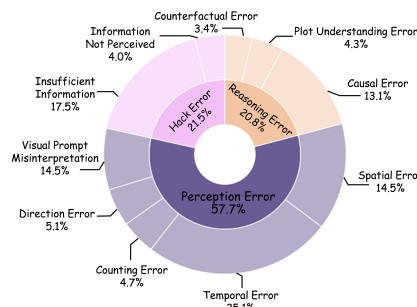

Figure 6: Distribution of Error Types.

- **Perception Error.** Provided with sufficient visual context, the model still produces errors due to deficiencies in its perception capabilities. Perception errors account for 57.7%, making them the most prevalent error type.

- **Reasoning Error.** The model demonstrates a range of logical reasoning failures, including plot understanding, counterfactual reasoning and causal errors. Some of these errors arises from deficiencies in perception, which further undermine the model's reasoning ability.

- **Hack Error.** Constrained by the sparse frame sampling strategy and the upper limit of the model's perceptual capability, the model fails to recognize sufficient visual semantics and consequently resorts to arbitrary guessing, which ultimately leads to erroneous predictions.

Analyzing and mitigating these errors is crucial for enhancing the performance of LVLMs in video visual prompt question understanding. This analysis provides an opportunity to target specific error types, thereby improving the model's overall capability.

## 5 CONCLUSION

In this study, we introduce V2P-Bench, a comprehensive benchmark for evaluating the video understanding capabilities of LVLMs through visual prompts in human–model interaction scenarios. Through a rigorous construction and evaluation process, V2P-Bench enables systematic evaluation and useful insights into the issues within the current models. Our experiments demonstrate three key findings: *1)* visual prompts outperform text prompts, substantially improving accuracy and interaction efficiency; *2)* a notable performance gap remains between LVLMs and human experts, especially in spatiotemporal understanding; and *3)* Hack Phenomena are prevalent, exacerbated by longer videos and sparser frame sampling. These results underscore the need for more robust evaluation protocols and model designs. We envision V2P-Bench as both a diagnostic tool and a roadmap for advancing LVLMs toward more reliable, human-aligned video understanding and interaction.

## 6 ACKNOWLEDGEMENTS

This work was supported by the Anhui Provincial Natural Science Foundation under Grant2108085UD12. We acknowledge the support of GPU cluster built by MCC Lab of Information Science and Technology Institution, USTC. The AI driven experiments, simulations and model training were performed on the robotic AI-Scientist platform of Chinese Academy of Sciences.

## 7 ETHICS STATEMENT

This work complies with ethical research standards in data collection, model training, and evaluation. All datasets used in this study are publicly available research datasets, collected and released under their respective licenses. No private or personally identifiable information (PII) was used.

## 8 REPRODUCIBILITY STATEMENT

We are committed to ensuring the reproducibility of our results. All prompts used during the evaluation are provided in the appendices. In addition, the datasets and evaluation scripts used in this paper have been publicly released.

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

APPENDIX OVERVIEW

- Section A: More Dataset Details.
- Section B: Dataset Scaling Up.
- Section C: More Experiments.
- Section D: Qualitative Examples.
- Section E: Implementation Details.
- Section F: Error Analysis.
- Section G: Application Scenarios for Visual Prompts.
- Section H: Discussion.
- Section I: Additional Results.
- Section J: Prompt Template.

# A   MORE DATASET DETAILS

## A.1   DATASET STATISTICS

In Table 1, we have already presented the main characteristics of V2P-Bench. Overall, the proposed V2P-Bench defines three main tasks and twelve dimensions, encompassing 980 unique videos and 1,172 QA pairs sourced from twelve existing video datasets, covering twenty video categories. The average duration of the videos is 19.0 minutes, which represents a wide range of video lengths, differing from most benchmarks. The format of the QA pairs is multiple-choice with 4 options. Below we introduce a more detailed analysis of our benchmark:

- **Wide distribution of durations.** Figure 7*(left)* shows the detailed duration distributions on V2P-Bench. We follow Video-MME (Fu et al., 2024) in categorizing video lengths into short (< 3 minutes), medium (3-30 minutes), and long videos (30-120 minutes), with respective proportions of 46.8%, 22.0%, and 31.2%.

- **Diverse video types and comprehensive tasks.** Figure 2*(left)* shows various datasets and categories on V2P-Bench. Figure 7*(right)* shows the detailed distribution of each dimension.

- **Diverse Targets and Visual Prompts.** Figure 3 shows various targets and visual prompts on V2P-Bench, benefiting from diverse video sources.

- **Comprehensive Shot Types.** V2P-Bench includes both continuous and transition videos, the latter of which significantly increases the difficulty of reference, implying that the model must perform temporal and spatial grounding in different scenes.

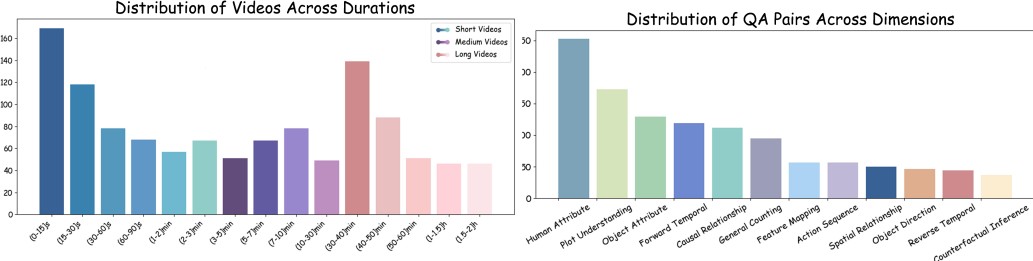

Figure 7: Distribution of video durations and dimensions.

## A.2 ELABORATION ON DIMENSIONS

Table 7 presents detailed information on the three main tasks and twelve dimensions of V2P-Bench.

Table 7: Our proposed three main tasks and twelve dimensions with explanation.

| Basic Perception | |
|---|---|
| **Object Attribute** | *This dimension evaluates the model's ability to perceive the visual and motion attributes of objects indicated by visual prompts, such as color, shape, position, and movement.* |
| **Human Attribute** | *This dimension evaluates the model's ability to recognize the actions and attributes of individuals indicated by visual prompts, such as their activities, clothing, and appearance.* |
| **Temporal Understanding** | |
| **Forward Temporal** | *This dimension assesses the model's ability to accurately locate the visual prompt and track subsequent events that follow the natural chronological order of the video.* |
| **Object Direction** | *This dimension examines the model's ability to perceive and interpret the motion trajectory of objects pointed by visual prompts, with a particular focus on movement direction.* |
| **Feature Mapping** | *This dimension examines the model's capability to extract distinctive features of objects indicated by visual prompts and consistently track them across the entire video.* |
| **Reverse Temporal** | *This dimension evaluates the model's capacity to comprehend the temporal structure of the video by identifying events that precede the visual prompt, demonstrating an understanding of temporal precedence.* |
| **Action Sequence** | *This dimension evaluates the model's ability to grasp the overall temporal flow of the video, particularly in understanding and reasoning about the temporal dynamics of multiple action sequences of individuals or objects, as indicated by visual prompts.* |
| **Spatial Relationship** | *This dimension assesses the model's ability to discern and comprehend the spatial relationships between instances highlighted by visual prompts within the video scene.* |
| **General Counting** | *This dimension evaluates the model's ability to perceive and accurately count repeated actions or objects within the video, as indicated by visual prompts, testing its capacity for detailed content understanding and comprehensive scene analysis.* |
| **High-level Reasoning** | |
| **Causal Relationship** | *This dimension assesses the model's ability to perceive the causal relationships between actions and events, identifying the underlying intentions of actions and the causes of subsequent events. The visual prompt points to the action executor.* |
| **Plot Understanding** | *This dimension examines the model's ability to analyze narrative progression and logically infer subsequent developments based on the given plot. The visual prompt executes the protagonist of the plot.* |
| **Counterfactual Inference** | *This dimension evaluates the model's ability to reason under hypothetical scenarios that deviate from the actual video content, with visual prompts guiding the deviation, assessing its capacity to infer potential outcomes based on counterfactual assumptions.* |

### A.3 ANNOTATION GUIDELINES

#### A.3.1 VISUAL PROMPT ANNOTATION

---

**Visual Prompt Annotation Guidelines**

**Interaction Constraint.** To remain consistent with real-world interactive applications, each QA pair is restricted to a single visual prompt frame. This design emphasizes the simplicity of user prompt creation and minimizes annotation burden. Therefore, annotators must select the most representative frame that anchors the key moment or location of the event.

**Visual Prompt Type.** Annotators should select the appropriate type of visual prompt according to the target and strictly follow the eight predefined categories.

**Uniqueness.** Visual prompts must precisely point to a specific object or region in the video, avoiding ambiguity or multiple referents.

**Consistency.** Visual prompts must strictly align with the question text, ensuring a one-to-one correspondence between the annotated target and the object referred to in the question, without mismatches or misleading references.

**Multi-target Differentiation.** When multiple prompts appear in the same frame, different shapes or colors should be used to clearly distinguish between targets.

---

#### A.3.2 QA PAIRS ANNOTATION

---

**QA Pairs Annotation Guidelines**

**Scenario Realism.** Questions must be based on the actual video content rather than hypothetical or fictional scenarios to ensure relevance to the real world and avoid fabricated or unrelated plots.

- Correct : Asking *What is the object held by the marked person?* when annotating a frame where a person raises a cup.
- Incorrect : *If he is holding a beer, what will he do next?* (overly speculative).

**Answer Uniqueness.** Questions should be concise and straightforward, avoiding ambiguous references or subjective judgments to ensure a unique answer.

- Correct : *What color of clothes is the target inside the rectangle wearing?*
- Incorrect : *What does this person look like?* (ambiguous and hard to standardize).

**Avoid Reliance on Common Sense.** Do not design questions that can be answered solely using common sense; ensure that the model must refer to the visual content to answer.

- Correct : *What color is the label of the target inside the ellipse?*
- Incorrect : *Do people usually use a knife to cut vegetables?* (answerable without watching the video).

**Fine-grained Observation and Diversity.** Questions should emphasize detailed observation and cover diverse objects (people, animals, tools, etc.), rather than being restricted to a specific target.

- Correct : *What color jersey is the circled target wearing?*
- Incorrect : *What activity is happening in the video?* (too broad).

**Dependence on Visual Prompts.** Questions must rely on visual prompts to be answered, which may require multiple targets to appear in the video, rather than inferring from context or common sense. Avoid questions unrelated to the prompts.

---

- Correct : *What object is the target pointed to by the arrow holding?*
- Incorrect : *Is the person in the video cooking?* (lacking visual prompt).

**Minimize Feature Descriptions.** Use only the visual prompts to refer to the target; avoid describing the target's appearance in words.

- Correct : When annotating a frame where an arrow points to a person sitting, ask: *What did the target do before sitting?*
- Incorrect : *What did the person wearing beige clothes and black pants do before sitting?* (contains appearance description).

**Concise Language and Distractor Design.** Questions and options should be kept concise to avoid noise from long descriptions and prevent models from learning biases unrelated to visuals. Options in multiple-choice questions should follow a consistent style, avoiding hints from length, tone, or format.

**Misleading yet incorrect distractors.** Besides common types (e.g., quantity substitution, causal reversal), annotators must carefully control the distractor's misleading nature to ensure answer uniqueness.

### A.3.3 DESIGN OF DISTRACTORS

Standards for Designing Distractors

**Quantity Substitution.** Replacing the correct numerical information (such as quantities, years, percentages, number of flags, etc.) with incorrect values.

**Tool / Object Replacement.** Swapping the actual tool used for another similar but different item—for example, writing hoe instead of rake, or switching the functions of a peeler and a knife.

**Role Reversal.** Exchanging the duties or action order of participants—for instance, changing "the cameraman cleans the table while the assistant kneads the dough" to the opposite distribution of tasks.

**Step / Sequence Inversion.** Reordering the correct procedure or omitting key steps—for example, writing the actions as "pass then plant" instead of the correct "plant then pass."

**Identity / Name Error.** Substituting the real person's name, nationality, or position with another—for example, writing Nathan instead of Eric, or labeling a Canadian athlete as British.

**Attribute Replacement.** Changing the attribute of clothing, objects, or chart lines to another common color—for example, describing black clothing as green.

**Orientation / Gesture Reversal.** Reversing directions or gestures—for example, writing "the right hand holds the cup" or "both hands hold it" instead of "the left hand holds the cup, and the right hand returns to the table."

**Emotion Substitution.** Replacing the true expression—such as "surprised" or "smiling"—with emotions like "angry," "sad," or "bored."

**Misplaced Cause-and-Effect.** Substituting the real reason with another seemingly plausible explanation—for example, the girl covering her ears is truly because "there are too many mice," but the distractor states "neighbors are renovating."

> **Location Replacement.** Changing the venue to a similar but different place—for instance, describing "a prison" as "a hospital" or "a hotel."
>
> **Relationship Misjudgment.** Describing friends as lovers, enemies, or strangers, or labeling a father-daughter pair as siblings, and so on.
>
> **Misattributed Result / Follow-up Action.** Giving an incorrect description of subsequent actions or impacts—for example, writing "the cameraman bows in thanks" instead of "waves and steps down."
>
> **Addition of Extraneous False Details.** Introducing nonexistent actions or objects—such as "handing the host a torch"—to create confusion through redundant information.

## A.4 QUALITY CONTROL

In Section 3.2.3, to ensure the quality of the dataset, we conducted a rigorous review of the annotated data after completion. The review process included Blind LLMs filtering and manual and rule-based review. Initially, we had 1,747 QA pairs. After applying Blind LLMs filtering, 1,653 pairs remained (94 filtered). Subsequently, we performed a comprehensive manual and rule-based review, resulting in a final count of 1,172 QA pairs (481 filtered). Table 8 shows some representative data examples from the post-processing step.

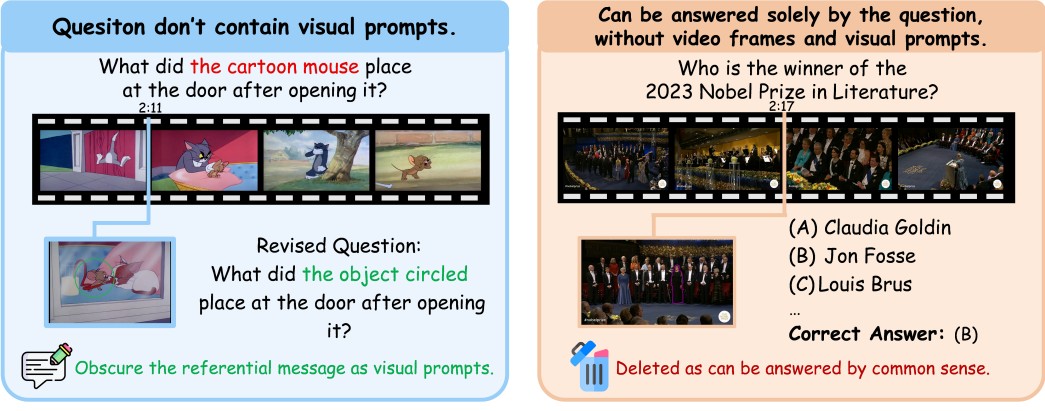

Figure 8: Representative data examples from the post-processing step.

## A.5 CERTIFICATE LENGTH

Following EgoSchema Mangalam et al. (2023), we compute the temporal certificate length and the temporal certificate ratio (the proportion of the minimal effective content duration relative to the total duration) based on the video length and task type in our dataset. The results are shown in Table 8.

Table 8: Certificate Length and Certificate Length Percentage across Different Video Durations.

|  | **Short** | | | **Medium** | | | **Long** | | |
|---|---|---|---|---|---|---|---|---|---|
|  | BP | TU | HR | BP | TU | HR | BP | TU | HR |
| **Certificate Length** | 1s | 31s | 27s | 1s | 204s | 169s | 1s | 1147s | 984s |
| **Certificate Length Percentage** | 1.9% | 61.4% | 52.9% | 0.2% | 44.7% | 37.2% | 0.003% | 35.3% | 30.3% |

The Basic Perception task is designed solely to assess a model's fundamental ability to perceive visual prompts, and thus it can be completed using a single prompted frame. In contrast, Temporal Understanding and High-level Reasoning focus on dynamic information and chronological relationships in videos, and on logical inference and higher-level judgment, respectively. As a result, both tasks require multiple video frames to be properly resolved.

### A.6 DATASET BIAS

Our benchmark is constructed from 12 publicly available video datasets, all of which were originally annotated with pure text-based QA pairs. The annotators of these datasets did not assume that "users will provide visual prompts" when creating the annotations, meaning that the data are naturally more aligned with text descriptions. The visual-prompt version of the benchmark is a rewrite we created on top of these original annotations. Therefore, from the perspective of data provenance: If any bias exists, it should favor pure text, not visual prompts.

Furthermore, to ensure a fair comparison between text prompts and visual prompts, we uniformly rewrote all text prompts rather than directly reusing the original textual annotations. Since the 12 constituent datasets vary widely in linguistic style, granularity, and descriptive conventions, directly comparing visual prompts with the original text would introduce additional bias. To avoid this, we rewrote all questions into a standardized text-prompt format and ensured the text prompts contained the same information as the visual prompts.

## B DATASET SCALING UP

The core value of a benchmark dataset lies in being high-quality, trustworthy, and diagnostic, rather than merely covering as much data as possible. To this end, we rely on trained human annotators instead of automated generation, ensuring that each question maintains a strict causal linkage between the video evidence, visual prompts, and the correct answer. Thus, the benchmark does not prioritize "scaling up data volume" as its primary goal.

On the other hand, to enable the construction of large-scale visual-prompt datasets and to enhance existing models' ability to understand visual prompts, we also provide a fully automated data-generation pipeline, including:

*1)* Using RAM++ Zhang et al. (2024a) to automatically extract potential target objects from video.

*2)* Applying SAM3 Carion et al. (2025) for cross-frame object tracking and mask propagation to obtain temporally consistent visual annotations.

*3)* Following automatic data-synthesis strategies from LLaVA-Video Zhang et al. (2024b) and ShareGPT4Video Chen et al. (2024c) to bind model-generated QA pairs to target regions, thus producing large quantities of high-quality video–visual-prompt supervision data.

## C MORE EXPERIMENTS

### C.1 CONSISTENCY ANALYSIS OF MCQ AND DIALOGUE TASKS

We adopt a MCQ format as MCQ-based evaluation is easy to automate, highly controllable, and effective at avoiding the ambiguity and vagueness that commonly arise in open-ended responses. This is also why existing mainstream video-understanding benchmarks (Video-MME Fu et al. (2024), MVBench Li et al. (2024c), EgoSchema Mangalam et al. (2023), LVBench Wang et al. (2024b) .etc)widely adopt a QA-style design. In contrast, free-form dialogue introduces substantial subjectivity and is difficult to score in a consistent and reproducible manner. Moreover, relying on an LLM-as-a-judge may introduce evaluation bias, further weakening the fairness, reliability, and comparability of the assessment. However, real human–model interaction scenarios are inherently dialogue tasks. To examine the extent to which a MCQ task can represent a dialogue task, we further analyze the consistency between the two. Specifically, we remove the multiple-choice constraints from the original questions, convert them into open-ended ones and re-evaluate the models using GPT-4o Hurst et al. (2024) and human annotators as the judge.

As shown in Table 9, under the open-ended setting, the Pearson consistency coefficient between MCQ and OE (GPT-4o Judge) reaches 0.95, and the coefficient between MCQ and OE (Human Judge) reaches 0.98. This demonstrates a strong consistency between QA-style evaluation and free-form dialogue, indicating that QA-based assessments can reliably reflect a model's capabilities in open-ended conversational scenarios.

Table 9: Consistency Evaluation Between MCQ and Open-Ended Tasks.

| Model | MCQ | OE (GPT-4o Judge) | OE (Human Judge) |
|---|---|---|---|
| Gemini-2.5-Pro | 69.8 | 48.3 | 51.6 |
| Qwen2.5-VL-7B | 48.1 | 30.2 | 33.7 |
| MiMo-VL-7B | 45.7 | 32.7 | 35.6 |

## C.2 HACK PHENOMENA IN OPEN-ENDED EVALUATION

Existing benchmarks commonly adopt the MCQ format because it is easy to evaluate and highly controllable, and avoids the ambiguity often seen in open-ended responses. Our experimental results show that models indeed exhibit such behavior: they tend to game the MCQ structure to obtain higher scores instead of performing true video comprehension. This finding suggests that, in addition to pursuing higher benchmark scores, future models should also prioritize genuine understanding and groundedness as key optimization objectives.

To examine whether open-ended generation helps reduce hack behaviors, we conducted two experiments in which we removed the multiple-choice constraints and answer options from the original prompts, converting them into open-ended question–answering tasks:

**Random Shuffling Experiment.** We randomly shuffle the videos and questions, meaning the model can no longer obtain the answer from the video content. If a model is sufficiently honest, it should refuse to answer all questions. We perform inference using Qwen2.5-VL-7B and MiMo-VL-7B. The Trigger Ratio denotes the proportion of cases in which the model refuses to answer. For the MCQ setting, we perform standard reasoning and count all responses that do not select any option; for the OE setting, we manually evaluate whether the model refused to respond. The results are shown in Table 10.

Table 10: Hack Phenomena under MCQ and Open-Ended Evaluation.

| Model | MCQ Trigger Ratio (%) | | | | OE Trigger Ratio (%) | | | |
|---|---|---|---|---|---|---|---|---|
| | Short | Medium | Long | Avg | Short | Medium | Long | Avg |
| Qwen2.5-VL-7B | 6.2 | 6.9 | 6.3 | 6.4 | 95.8 | 96.8 | 96.1 | 96.7 |
| MiMo-VL-7B | 3.6 | 4.2 | 4.0 | 3.9 | 92.2 | 94.6 | 94.8 | 93.6 |

We observe that under the MCQ setting, Qwen2.5-VL-7B and MiMo-VL-7B exhibit refusal rates of only 6.4% and 3.9%, respectively. This indicates that even when the video and question are completely mismatched, the models still select an option in the vast majority of cases, and the trigger rates remain almost identical across different video lengths. In contrast, under the OE setting, the refusal rates rise sharply to 96.7% and 93.6%, again showing little variation across video lengths. These results clearly demonstrate that open-ended generation effectively mitigates hack behaviors to a large extent.

**Model Performance on V2P-Bench.** Besides,we follow the same setup as in the random shuffling experiment, with the only difference being that we use the original benchmark data rather than the shuffled version. The experimental results are shown in Table 11.

Table 11: Hack Rates under MCQ and Open-Ended Evaluation.

| Model | MCQ Hack Ratio (%) | | | | OE Hack Ratio (%) | | | |
|---|---|---|---|---|---|---|---|---|
| | Short | Medium | Long | Avg | Short | Medium | Long | Avg |
| Qwen2.5-VL-7B | 8.1 | 11.5 | 14.4 | 10.2 | 2.1 | 2.6 | 3.4 | 2.7 |
| MiMo-VL-7B | 5.3 | 7.2 | 7.4 | 6.2 | 1.7 | 2.3 | 2.8 | 2.2 |

We observe that after converting the MCQ task into an open-ended question–answering task, the Hack Ratios of Qwen2.5-VL-7B and MiMo-VL-7B decrease by 7.5% and 4.0%, respectively. This further demonstrates that open-ended generation can substantially mitigate hack behaviors.

## C.3 Impact of Sampling Frame Rates

We evaluate Qwen2.5-VL-7B across different task types and video durations to investigate how varying sampling frame rates affect model performance. Here, 1 frame indicates that only the visual-prompt frame is provided to the model. The results are shown in Table 12.

Table 12: Performance under Different Sampling Frame Rates across Video Durations.

| Model | Frames | Short | | | Medium | | | Long | | |
|---|---|---|---|---|---|---|---|---|---|---|
| | | BP | TU | HR | BP | TU | HR | BP | TU | HR |
| | 1 | 62.1 | 28.6 | 29.8 | 64.3 | 30.4 | 31.1 | 47.6 | 27.4 | 33.2 |
| Qwen2.5-VL-7B | 8 | 61.8 | 38.3 | 43.0 | 59.1 | 43.6 | 44.2 | 44.8 | 41.7 | 45.7 |
| | 64 | 60.8 | 40.1 | 46.8 | 56.1 | 47.9 | 51.8 | 43.1 | 41.7 | 50.0 |

We observe that for the BP task, which relies solely on single-frame information, increasing the sampling frame rate actually leads to a certain degree of performance degradation. This is because more frames enlarge the temporal search space, making it harder for the model to accurately locate the visual prompt frame. In contrast, TU and HR tasks inherently require multi-frame temporal information. As the frame rate increases, the model gains access to richer dynamic cues, resulting in a clear performance improvement that eventually saturates.

## D Qualitative Examples

### D.1 Comparison with INST-IT

Unlike V2P-Bench, INST-IT dataset contains visual prompt annotations on every frame of the video, as shown in Figure 9 *(left)*, which is both unrealistic and practically unachievable in real human-model interaction scenarios.

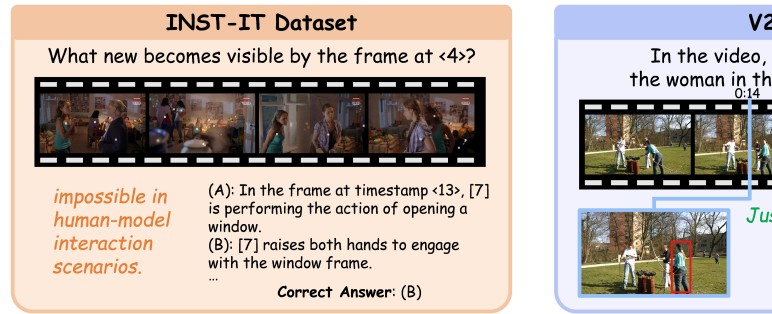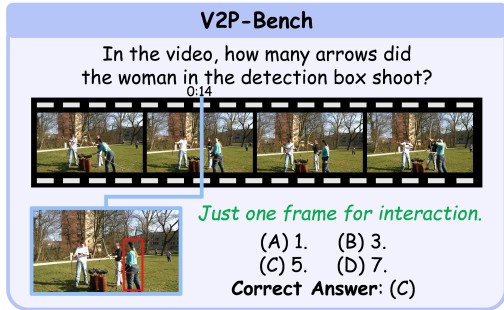

Figure 9: Comparison of INST-IT-Dataset and V2P-Bench.

## E Implementation Details

### E.1 Human Exam and Blind LLMs Answering.

**Human Exam**. For the human evaluation, all questions were assigned to three human experts. To mitigate the risk of data leakage, we took special care to ensure that the experts involved in the evaluation had never participated in the annotation process. The human experts were instructed to watch the videos along with the visual prompt frames and to respond by selecting only the letter corresponding to their chosen option. The results were recorded in a JSON file. The evaluation was scored using a script, maintaining consistency with the model evaluation process.

**Blind LLMs Answering.** For the blind answering task, we report the performance of three models: GPT-4o, Gemini-2.5-Pro, and Qwen2.5-VL. Although all three models are in fact LVLMs, we provide them only with the textual QA pairs, so that their visual encoders remain inactive. The models are

prompted to output "Z" honestly instead of selecting an option whenever the QA pair requires access to visual context, thereby preventing the models from guessing.

### E.2 COMPARISON OF TEXT AND VISUAL PROMPTS FOR HUMANS AND MODELS.

**Text prompt evaluation.** After completing the annotation of the visual-prompt dataset, we uniformly rewrote all text prompts, rather than directly reusing the original textual annotations. The twelve source datasets differ significantly in linguistic style, granularity, and description patterns; therefore, directly comparing their original questions would introduce additional bias. To ensure fairness, we rewrote all questions into a standardized text-prompt format and guaranteed that the amount of information conveyed matched that of the visual-prompt version.
Concretely, we converted the visual-prompt references in each QA pair into natural-language descriptions of the target's appearance. For example, a question such as "What does the arrow-pointed person do after getting off the car?" was rewritten as "What does the person weaormuing a black suit and black hat do after getting off the car?" In contrast, in the original text-only dataset, the same question appeared as "What does the main character in the video do after getting off the car?", a formulation that requires the model to first understand the video content and identify the main character. This would be unfair when comparing text prompts to visual prompts, since the latter already explicitly localize the target. Therefore, we performed a unified rewriting of all text prompts to eliminate this bias and ensure a fair comparison between text-based and visual-prompt–based evaluation.
To guarantee fairness in the comparison, we also include the original video frame corresponding to the visual prompt (without any visual prompts) later in the video, ensuring that in both experimental settings, the model receives an identical amount of visual context. Finally, we adopt the same inference and evaluation procedures as in the visual prompt setting, ensuring comparability of results and enabling a rigorous assessment of model performance in pure text QA tasks.

**User experience study.** In fact, we reused the Gradio-based annotation interface originally built for constructing the dataset, so the volunteers operated in the same environment as the annotators. The UI supports displaying, playing, and pausing the video at the top; when the video is paused, clicking again enlarges the frame and launches MS Paint, allowing users to draw visual prompts directly on the frame. The UI also displays the original QA pair and allows the annotator to input a new question. After the participant finishes formulating the question, the video, the visual prompt frame, and the question are sent to the Gemini API to obtain a response. We slightly improved the system to display the model's reply within the UI.
To avoid possible biases or preferences arising from annotators who were familiar with the dataset's purpose, we recruited volunteers to conduct the user study. Before formally participating, each volunteer received training from one of the annotators and completed a guided interaction session. Only after this training were the volunteers allowed to proceed with the actual experiment.

## F  Error Analysis

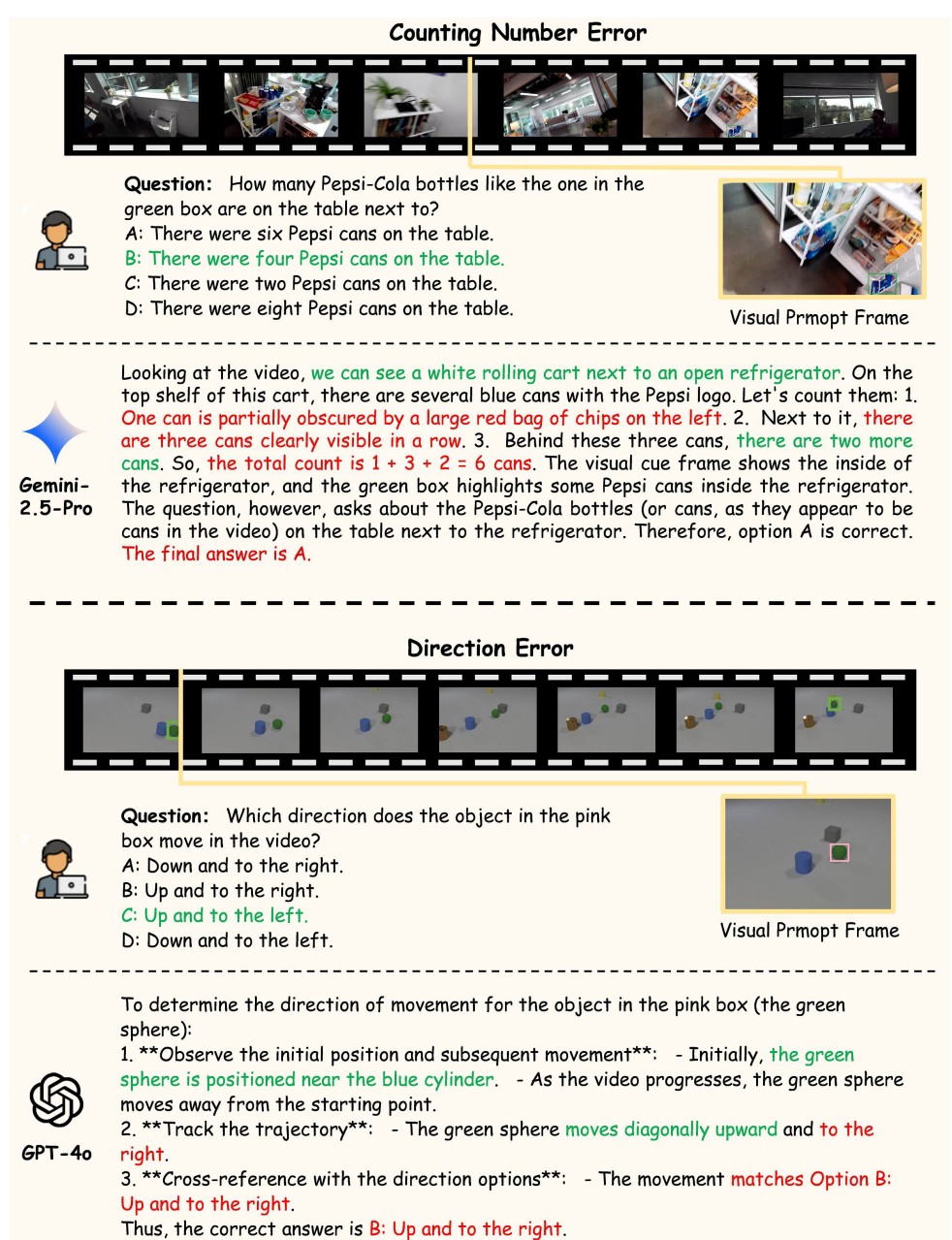

Figure 10: Typical failure modes of the model regarding Perception Errors.

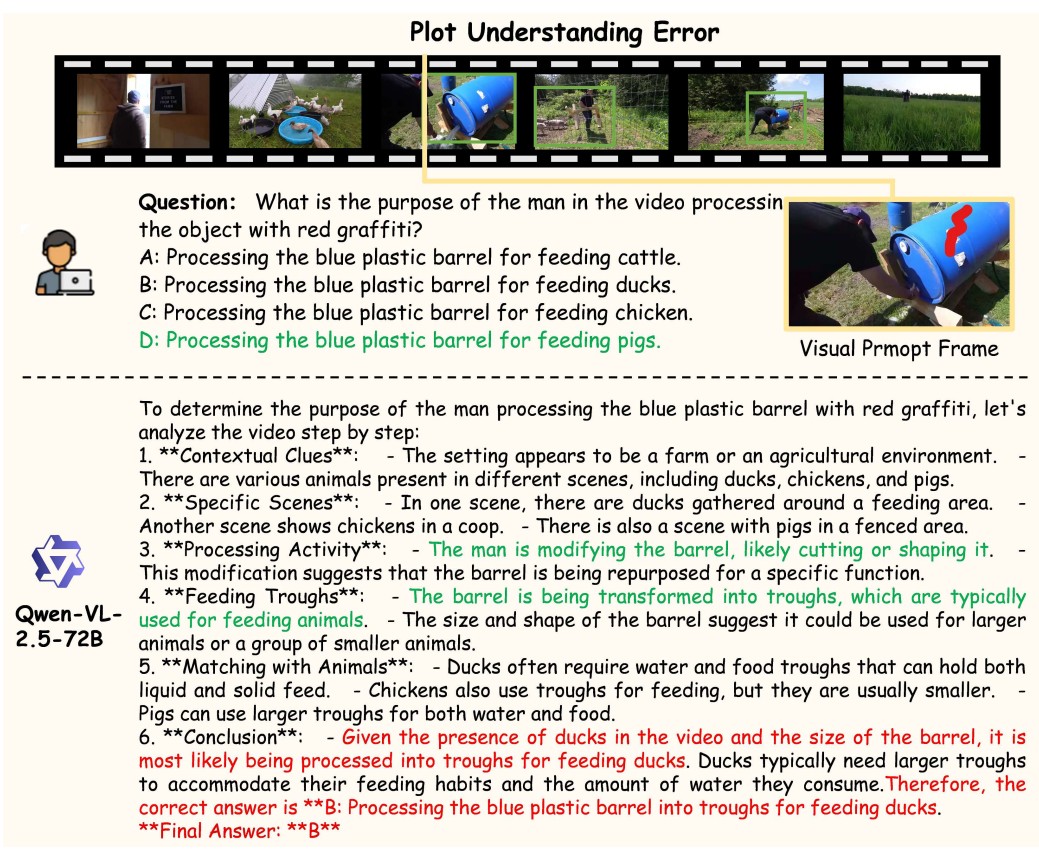

Figure 11: Typical failure modes of the model regarding Reasoning Errors.

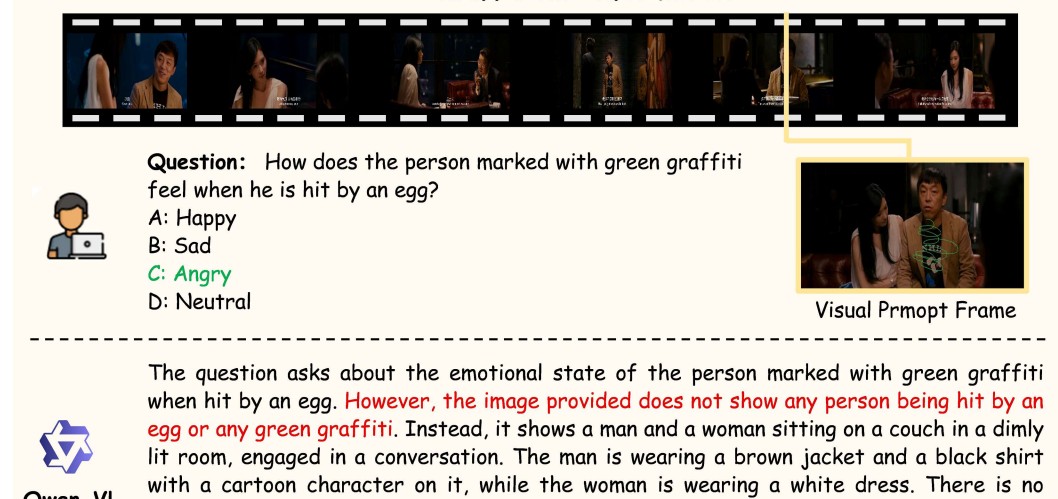

Figure 12: Typical failure modes of the model regarding Insufficient Information Errors.

## G  APPLICATION SCENARIOS FOR VISUAL PROMPTS

In real-world scenarios, the video modality demonstrates greater applicability and value, specifically including but not limited to the following situations:

- **Interaction with Mobile Devices.** Users can watch videos on mobile devices such as smartphones and computers, easily creating visual prompts with their fingers or a mouse. This functionality has been successfully implemented in systems like GPT-4o and Video Refer, allowing users to interact with video content more intuitively, enhancing their viewing experience.

- **Smart Device Wearables.** Imagine users wearing advanced smart devices (like Apple Vision Pro), immersed in a sea of video content. These users could generate visual prompts through natural gestures. It undoubtedly offers limitless possibilities for future interaction methods, significantly enhancing the immersive experience of watching videos.

- **Interaction with Smart Robots.** Consider the interaction between users and physical smart robot terminals equipped with visual prompt screens. It will revolutionize the way we communicate with smart technology, making interactions more vivid, engaging, and intuitive.

In these diverse scenarios, users can interact smoothly with local or cloud-based LVLMs based on video content. Developers only need to adjust the front-end system to efficiently capture video frames and visual prompts to achieve this goal. Such integration will create a more immersive experience for users, transforming video watching from passive observation into an engaging and interactive journey.

## H  DISCUSSION

**Limitation.** Although our V2P-Bench comprehensively evaluates the capabilities of LVLMs in video-language understanding with visual prompts for multimodal human-model interaction, it only focuses on visual and textual inputs, lacking audio input, and supports evaluations only on offline videos, which leaves a gap compared to the ultimate form of multimodal human-model interaction in real world. we plan to develop V2P-Bench v2, which will support all types of video understanding, incorporate full-modality inputs, and enable the evaluation of multi-turn dialogue and interruptible interactions.

**Broader Impact.** V2P-Bench has built a comprehensive visual prompt dataset for evaluating video visual prompt question answering in the multimodal domain, which will help more thoroughly validate the video understanding capabilities of large visual language models and enhance their performance in the field of video understanding. We have released the dataset, evaluation code, and leaderboard.

## I  ADDITIONAL RESULTS

### I.1  RESULTS ACROSS VISUAL PROMPTS

See Table 13.

### I.2  RESULTS ACROSS FRAME RATES

See Table 14 to Table 22.

### I.3  HACK PHENOMENA ON OTHER BENCHMARKS

See Figure 13.

## J    PROMPT TEMPLATE

---

**Prompt template for model inference**

**System prompt:**

First, you will receive a series of images sampled at regular intervals from a video. Then, you will receive a visual prompt frame, which is screenshot from the same video and have been manually annotated with visual prompts. Your task is to answer the question based on the video and visual prompt frame.
Select the best option that accurately addresses the question. Give only your option letter, no other words.

**User prompt:**

```
<video> <vp_frame> <question>
```

---

**Prompt template for the blind answering task**

**System prompt:**

You will receive a question and four options. Please respond based on the question.
Select the best option that accurately addresses the question. Give only your option letter, no other words.
If the question cannot be answered, output Z.

**User prompt:**

```
<question>
```

---

**Prompt template for model inference without hack**

**System prompt:**

First, you will receive a series of images sampled at regular intervals from a video. Then, you will receive a visual prompt frame, which is screenshot from the same video and have been manually annotated with visual prompts. Your task is to answer the question based on the video and visual prompt frame.
Select the best option that accurately addresses the question. Give only your option letter, no other words.
If the video does not provide sufficient information, or if none of the options is correct, don't try to make up an answer but output Z directly.

**User prompt:**

```
<video> <vp_frame> <question>
```

---

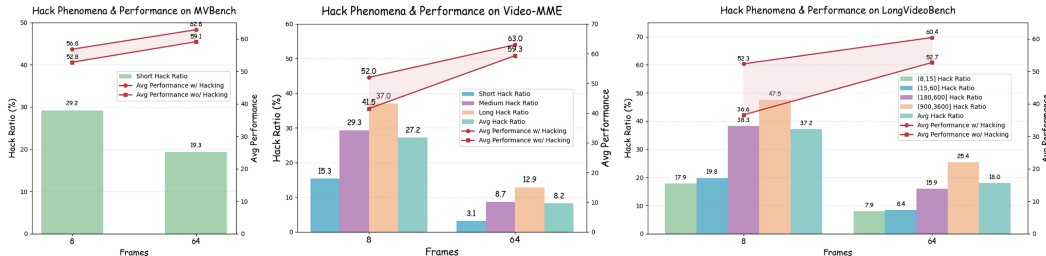

Figure 13: Hack Phenomena and Performance of Qwen2.5-VL-7B on MVBench, Video-MME and LongVideoBench.

Table 13: Evaluation results on V2P-Bench across visual prompt types.

| Method | Rectangle | Mask | Ellipse | Triangle | Scribble | Point | Arrow | SoM | Avg |
|---|---|---|---|---|---|---|---|---|---|
| Human Performance | 86.6 | 88.1 | 92.5 | 86.1 | 86.9 | 89.7 | 90.2 | 86.3 | 88.3 |
| *Closed-source Models* | | | | | | | | | |
| o1 | 74.4 | 70.1 | 70.3 | 68.9 | 71.9 | 74.5 | 72.1 | 68.3 | 71.8 |
| GPT-4o | 68.5 | 63.6 | 65.4 | 62.3 | 66.0 | 63.2 | 69.8 | 59.6 | 65.4 |
| Gemini-2.5-Pro | 73.2 | 63.6 | 71.4 | 65.6 | 69.5 | 72.4 | 62.8 | 66.3 | 69.8 |
| *Open-source Models* | | | | | | | | | |
| LLaVA-NeXT-7B | 44.3 | 40.2 | 47.9 | 49.1 | 48.3 | 45.4 | 45.1 | 47.7 | 46.0 |
| LLaVA-NeXT-INST-IT-7B | 42.2 | 41.5 | 45.0 | 45.6 | 45.3 | 43.7 | 44.2 | 62.7 | 46.3 |
| LLaVA-OV-7B | 54.7 | 48.1 | 49.2 | 59.0 | 44.9 | 72.1 | 52.7 | 52.9 | 52.8 |
| LLaVA-OV-72B | 55.1 | 54.5 | 54.1 | 55.7 | 58.2 | 60.5 | 56.7 | 54.8 | 56.7 |
| InternVL3-8B | 62.6 | 57.1 | 61.1 | 62.3 | 65.3 | 55.8 | 59.6 | 65.4 | 61.7 |
| mPLUG-Owl3-7B | 51.6 | 55.8 | 47.6 | 52.5 | 49.0 | 53.5 | 59.1 | 51.9 | 52.6 |
| LLaVA-Video-7B | 50.4 | 54.5 | 53.5 | 55.7 | 62.2 | 58.1 | 56.2 | 54.8 | 54.8 |
| LLaVA-Video-72B | 59.1 | 59.7 | 57.3 | 49.2 | 67.3 | 55.8 | 58.6 | 56.7 | 58.6 |
| MiniCPM-V 2.6-8B | 56.3 | 48.1 | 53.5 | 65.6 | 51.0 | 53.5 | 57.6 | 53.8 | 55.3 |
| Qwen2.5-VL-7B | 47.6 | 55.8 | 42.2 | 42.6 | 48.0 | 48.8 | 50.2 | 51.9 | 48.1 |
| Qwen2.5-VL-72B | 63.4 | 61.0 | 57.8 | 54.1 | 55.1 | 58.1 | 61.1 | 61.5 | 59.8 |
| MiMo-VL-7B | 49.6 | 51.9 | 55.1 | 55.7 | 50.0 | 48.8 | 56.2 | 61.5 | 53.8 |

Table 14: Results on LLaVA-OneVision-7B with different frame-rate sampling strategies.

| Model | Frames | Dimension | | | | | | | | | | | Duration | | | Avg |
|---|---|---|---|---|---|---|---|---|---|---|---|---|---|---|---|---|
| | | OA | HA | OD | FM | CR | PU | CI | FT | RT | AS | SR | GC | Short | Medium | Long | |
| LLaVA-OV-7B | 4 | 49.6 | 52.5 | 30.4 | 45.1 | 51.4 | 56.8 | 33.3 | 47.4 | 36.7 | 45.3 | 57.2 | 35.1 | 51.1 | 48.1 | 40.3 | 48.0 |
| | 8 | 51.3 | 51.2 | 32.6 | 45.1 | 56.2 | 59.1 | 41.0 | 47.4 | 37.8 | 48.4 | 55.2 | 43.2 | 52.4 | 50.0 | 41.2 | 49.2 |
| | 16 | 58.8 | 53.0 | 28.3 | 45.1 | 56.2 | 50.0 | 35.9 | 34.2 | 37.8 | 55.8 | 58.6 | 43.2 | 52.0 | 53.7 | 43.6 | 50.4 |
| | 32 | 56.3 | 53.0 | 28.3 | 43.1 | 58.1 | 56.8 | 41.0 | 36.8 | 38.9 | 50.6 | 64.1 | 43.2 | 52.0 | 56.9 | 44.0 | 51.2 |
| | 64 | 56.3 | 54.4 | 28.3 | 47.1 | 60.0 | 59.1 | 41.0 | 42.1 | 41.1 | 55.8 | 63.4 | 43.2 | 51.9 | 63.4 | 45.3 | 52.7 |
| | 128 | 57.1 | 52.1 | 28.3 | 47.1 | 63.8 | 59.1 | 41.0 | 42.1 | 35.6 | 63.2 | 62.8 | 43.2 | 51.3 | 63.0 | 47.3 | 52.8 |

Table 15: Results on LLaVA-OneVision-72B with different frame-rate sampling strategies.

| Model | Frames | Dimension | | | | | | | | | | | Duration | | | Avg |
|---|---|---|---|---|---|---|---|---|---|---|---|---|---|---|---|---|
| | | OA | HA | OD | FM | CR | PU | CI | FT | RT | AS | SR | GC | Short | Medium | Long | |
| LLaVA-OV-72B | 4 | 48.7 | 55.8 | 37.0 | 47.1 | 55.2 | 50.0 | 38.5 | 57.9 | 33.3 | 60.0 | 60.0 | 32.4 | 50.6 | 57.4 | 46.1 | 51.0 |
| | 8 | 58.0 | 55.3 | 39.1 | 47.1 | 57.1 | 43.2 | 33.3 | 52.6 | 35.6 | 62.1 | 57.9 | 45.9 | 52.6 | 59.7 | 44.4 | 52.1 |
| | 16 | 62.2 | 57.6 | 30.4 | 43.1 | 59.0 | 38.6 | 35.9 | 47.4 | 37.8 | 62.1 | 66.2 | 45.9 | 52.4 | 63.9 | 48.1 | 53.8 |
| | 32 | 62.2 | 60.8 | 30.4 | 45.1 | 63.8 | 45.5 | 41.0 | 52.6 | 35.6 | 62.1 | 67.6 | 45.9 | 54.9 | 65.3 | 49.4 | 55.8 |
| | 64 | 65.5 | 60.4 | 30.4 | 45.1 | 61.9 | 43.2 | 38.5 | 47.4 | 43.3 | 65.3 | 66.2 | 45.9 | 53.8 | 71.8 | 48.1 | 56.2 |
| | 128 | 65.5 | 59.9 | 34.7 | 47.0 | 63.8 | 43.2 | 38.5 | 50.0 | 41.1 | 66.3 | 66.9 | 45.9 | 54.5 | 70.4 | 49.0 | 56.7 |

Table 16: Results on InternVL3-8B with different frame-rate sampling strategies.

| Model | Frames | Dimension | | | | | | | | | | | Duration | | | Avg |
|---|---|---|---|---|---|---|---|---|---|---|---|---|---|---|---|---|
| | | OA | HA | OD | FM | CR | PU | CI | FT | RT | AS | SR | GC | Short | Medium | Long | |
| InternVL3-8B | 4 | 66.4 | 65.0 | 28.3 | 56.9 | 51.4 | 59.1 | 28.2 | 52.6 | 34.4 | 57.9 | 63.4 | 64.9 | 56.8 | 58.8 | 51.9 | 56.0 |
| | 8 | 69.7 | 67.7 | 39.1 | 58.8 | 62.9 | 59.1 | 30.8 | 44.7 | 33.3 | 52.6 | 68.3 | 70.3 | 59.6 | 65.3 | 51.4 | 58.9 |
| | 16 | 69.7 | 67.7 | 37.0 | 54.9 | 61.9 | 63.6 | 35.9 | 52.6 | 37.8 | 60.0 | 68.3 | 67.6 | 60.3 | 68.5 | 52.3 | 60.1 |
| | 32 | 69.7 | 67.3 | 39.1 | 52.9 | 63.8 | 63.6 | 43.6 | 50.0 | 40.0 | 56.8 | 70.3 | 62.2 | 59.8 | 68.1 | 55.1 | 60.4 |
| | 64 | 68.1 | 67.7 | 34.8 | 62.7 | 56.2 | 63.6 | 35.9 | 57.9 | 37.8 | 63.2 | 74.5 | 70.3 | 58.2 | 71.8 | 58.4 | 61.1 |
| | 128 | 73.9 | 69.1 | 39.1 | 60.8 | 58.1 | 65.9 | 41.0 | 52.6 | 41.1 | 61.1 | 69.7 | 64.9 | 61.7 | 68.5 | 55.6 | 61.7 |

Table 17: Results on mPLUG-Owl3-7B with different frame-rate sampling strategies.

| Model | Frames | Dimension | | | | | | | | | | | | Duration | | | Avg |
|---|---|---|---|---|---|---|---|---|---|---|---|---|---|---|---|---|---|
| | | OA | HA | OD | FM | CR | PU | CI | FT | RT | AS | SR | GC | Short | Medium | Long | |
| mPLUG-Owl3-7B | 4 | 61.3 | 53.5 | 32.6 | 41.2 | 53.3 | 50.0 | 48.7 | 55.3 | 26.7 | 51.6 | 53.8 | 37.8 | 51.7 | 50.5 | 43.6 | 49.5 |
| | 8 | 62.2 | 54.4 | 34.8 | 51.0 | 50.5 | 43.2 | 41.0 | 50.0 | 23.3 | 52.6 | 57.9 | 37.8 | 50.3 | 54.2 | 44.4 | 49.7 |
| | 16 | 59.7 | 51.6 | 30.4 | 51.0 | 54.3 | 45.5 | 48.7 | 55.3 | 31.1 | 54.7 | 60.0 | 40.5 | 51.3 | 56.9 | 44.4 | 50.9 |
| | 32 | 60.5 | 55.3 | 32.6 | 45.1 | 56.2 | 45.5 | 48.7 | 50.0 | 28.9 | 54.7 | 61.4 | 35.1 | 51.7 | 58.8 | 44.0 | 51.4 |
| | 64 | 55.5 | 54.8 | 34.8 | 54.9 | 60.0 | 50.0 | 46.2 | 50.0 | 33.3 | 58.9 | 58.6 | 35.1 | 52.2 | 59.3 | 45.7 | 52.1 |
| | 128 | 61.3 | 54.4 | 28.3 | 49.0 | 60.0 | 50.0 | 51.3 | 60.5 | 34.4 | 55.8 | 58.6 | 37.8 | 52.2 | 62.5 | 44.9 | 52.6 |

Table 18: Results on LLaVA-Video-7B with different frame-rate sampling strategies.

| Model | Frames | Dimension | | | | | | | | | | | | Duration | | | Avg |
|---|---|---|---|---|---|---|---|---|---|---|---|---|---|---|---|---|---|
| | | OA | HA | OD | FM | CR | PU | CI | FT | RT | AS | SR | GC | Short | Medium | Long | |
| LLaVA-Video-7B | 4 | 48.7 | 56.2 | 32.6 | 51.0 | 55.2 | 54.5 | 35.9 | 52.6 | 38.9 | 51.6 | 55.9 | 35.1 | 49.6 | 56.9 | 45.7 | 50.2 |
| | 8 | 55.5 | 54.8 | 30.4 | 49.0 | 54.3 | 59.1 | 38.5 | 55.3 | 43.3 | 54.7 | 53.1 | 37.8 | 50.6 | 57.4 | 46.9 | 51.2 |
| | 16 | 61.3 | 57.6 | 32.6 | 49.0 | 57.1 | 54.5 | 38.5 | 52.6 | 41.1 | 56.8 | 59.3 | 43.2 | 53.4 | 62.0 | 46.5 | 53.6 |
| | 32 | 58.8 | 58.5 | 34.8 | 52.9 | 58.1 | 56.8 | 51.3 | 55.3 | 42.2 | 56.8 | 55.2 | 40.5 | 54.3 | 63.4 | 46.9 | 54.0 |
| | 64 | 58.0 | 59.4 | 32.6 | 47.1 | 61.9 | 56.8 | 41.0 | 52.6 | 44.4 | 54.7 | 61.4 | 40.5 | 54.3 | 62.5 | 47.7 | 54.5 |
| | 128 | 60.5 | 58.1 | 37.0 | 49.0 | 62.9 | 54.5 | 41.0 | 52.6 | 48.9 | 57.9 | 56.6 | 40.5 | 54.1 | 65.7 | 46.5 | 54.8 |

Table 19: Results on LLaVA-Video-72B with different frame-rate sampling strategies.

| Model | Frames | Dimension | | | | | | | | | | | | Duration | | | Avg |
|---|---|---|---|---|---|---|---|---|---|---|---|---|---|---|---|---|---|
| | | OA | HA | OD | FM | CR | PU | CI | FT | RT | AS | SR | GC | Short | Medium | Long | |
| LLaVA-Video-72B | 4 | 52.9 | 54.8 | 43.5 | 54.9 | 55.2 | 56.8 | 43.6 | 57.9 | 32.2 | 61.1 | 64.8 | 54.1 | 55.2 | 58.8 | 46.5 | 53.9 |
| | 8 | 58.0 | 54.4 | 34.8 | 52.9 | 61.0 | 56.8 | 41.0 | 47.4 | 43.3 | 63.2 | 61.4 | 56.8 | 55.2 | 62.5 | 46.9 | 54.8 |
| | 16 | 61.3 | 55.8 | 34.8 | 51.0 | 60.0 | 56.8 | 43.6 | 52.6 | 40.0 | 65.3 | 68.3 | 62.2 | 56.4 | 65.3 | 49.4 | 56.6 |
| | 32 | 65.5 | 58.5 | 32.6 | 49.0 | 57.1 | 50.0 | 46.2 | 50.0 | 44.4 | 65.3 | 71.0 | 54.1 | 55.9 | 67.6 | 51.9 | 57.4 |
| | 64 | 61.3 | 59.4 | 23.9 | 51.0 | 64.8 | 50.0 | 48.7 | 50.0 | 44.4 | 69.5 | 72.4 | 45.9 | 55.7 | 69.0 | 53.5 | 58.0 |
| | 128 | 62.2 | 60.8 | 30.4 | 54.9 | 61.0 | 54.5 | 43.6 | 47.4 | 42.2 | 70.5 | 71.0 | 59.5 | 57.5 | 66.2 | 54.3 | 58.6 |

Table 20: Results on MiniCPM-V 2.6 with different frame-rate sampling strategies.

| Model | Frames | Dimension | | | | | | | | | | | | Duration | | | Avg |
|---|---|---|---|---|---|---|---|---|---|---|---|---|---|---|---|---|---|
| | | OA | HA | OD | FM | CR | PU | CI | FT | RT | AS | SR | GC | Short | Medium | Long | |
| MiniCPM-V 2.6 | 4 | 59.7 | 54.8 | 28.3 | 41.2 | 42.9 | 52.3 | 28.2 | 52.6 | 33.3 | 51.6 | 59.3 | 40.5 | 50.3 | 51.4 | 44.0 | 49.0 |
| | 8 | 63.0 | 55.3 | 26.1 | 33.3 | 52.4 | 43.2 | 30.8 | 44.7 | 34.4 | 44.2 | 64.1 | 40.5 | 49.9 | 55.1 | 43.6 | 49.5 |
| | 16 | 60.5 | 59.0 | 26.1 | 51.0 | 58.1 | 52.3 | 30.8 | 47.4 | 35.6 | 55.8 | 61.4 | 43.2 | 52.9 | 63.0 | 44.0 | 52.9 |
| | 32 | 65.5 | 57.1 | 28.3 | 49.0 | 58.1 | 50.0 | 33.3 | 50.0 | 43.3 | 55.8 | 60.7 | 45.9 | 54.5 | 61.6 | 45.3 | 53.8 |
| | 64 | 67.2 | 62.2 | 26.1 | 52.9 | 57.1 | 43.2 | 28.2 | 50.0 | 40.0 | 58.9 | 64.8 | 45.9 | 54.3 | 65.7 | 47.7 | 55.2 |
| | 128 | 68.9 | 59.4 | 26.1 | 56.9 | 58.1 | 50.0 | 33.3 | 50.0 | 34.4 | 57.9 | 67.6 | 43.2 | 53.3 | 66.2 | 50.2 | 55.3 |

Table 21: Results on Qwen2.5-VL-7B with different frame-rate sampling strategies.

| Model | Frames | Dimension | | | | | | | | | | | | Duration | | | Avg |
|---|---|---|---|---|---|---|---|---|---|---|---|---|---|---|---|---|---|
| | | OA | HA | OD | FM | CR | PU | CI | FT | RT | AS | SR | GC | Short | Medium | Long | |
| Qwen2.5-VL-7B | 4 | 59.7 | 51.6 | 21.7 | 49.0 | 46.7 | 43.2 | 33.3 | 44.7 | 35.6 | 41.1 | 48.3 | 45.9 | 48.0 | 48.6 | 39.9 | 46.2 |
| | 8 | 62.2 | 56.2 | 17.4 | 35.3 | 45.7 | 43.2 | 53.8 | 57.9 | 36.7 | 41.1 | 47.6 | 35.1 | 47.6 | 49.1 | 43.2 | 46.9 |
| | 16 | 56.3 | 54.8 | 26.1 | 47.1 | 47.6 | 52.3 | 25.6 | 52.6 | 36.7 | 44.2 | 49.0 | 40.5 | 49.2 | 48.1 | 42.4 | 47.4 |
| | 32 | 56.3 | 53.9 | 26.1 | 43.1 | 52.4 | 56.8 | 38.5 | 55.3 | 32.2 | 43.2 | 46.2 | 43.2 | 50.1 | 48.6 | 40.3 | 47.5 |
| | 64 | 62.2 | 53.9 | 19.6 | 41.2 | 47.6 | 50.0 | 38.5 | 57.9 | 33.3 | 44.2 | 49.7 | 40.5 | 48.7 | 52.3 | 43.2 | 48.1 |
| | 128 | 60.5 | 53.7 | 17.4 | 45.1 | 47.6 | 40.9 | 48.7 | 52.6 | 32.2 | 47.4 | 50.3 | 35.1 | 48.0 | 53.2 | 43.6 | 48.1 |

Table 22: Results on Qwen2.5-VL-72B with different frame-rate sampling strategies.

| Model | Frames | Dimension | | | | | | | | | | | | Duration | | | Avg |
|---|---|---|---|---|---|---|---|---|---|---|---|---|---|---|---|---|---|
| | | OA | HA | OD | FM | CR | PU | CI | FT | RT | AS | SR | GC | Short | Medium | Long | |
| Qwen2.5-VL-72B | 4 | 64.7 | 65.0 | 28.3 | 56.9 | 41.0 | 40.9 | 30.8 | 55.3 | 46.7 | 50.5 | 59.3 | 29.7 | 55.4 | 55.6 | 44.0 | 52.7 |
| | 8 | 65.5 | 62.2 | 30.4 | 49.0 | 42.9 | 52.3 | 38.5 | 55.3 | 37.8 | 54.7 | 59.3 | 40.5 | 55.7 | 52.8 | 46.5 | 52.9 |
| | 16 | 65.5 | 66.4 | 39.1 | 51.0 | 45.7 | 50.0 | 46.2 | 55.3 | 40.0 | 55.8 | 61.4 | 43.2 | 58.0 | 57.9 | 47.3 | 55.5 |
| | 32 | 67.2 | 68.2 | 47.8 | 47.1 | 51.4 | 52.3 | 53.8 | 52.6 | 43.3 | 57.9 | 61.4 | 51.4 | 61.6 | 58.8 | 48.6 | 57.9 |
| | 64 | 69.7 | 68.2 | 39.1 | 54.9 | 47.6 | 50.0 | 51.3 | 50.0 | 44.4 | 63.2 | 68.3 | 48.6 | 61.0 | 62.0 | 51.4 | 59.0 |
| | 128 | 69.7 | 72.4 | 43.5 | 52.9 | 49.5 | 59.1 | 53.8 | 55.3 | 44.4 | 57.9 | 64.1 | 51.4 | 62.4 | 63.9 | 50.2 | 59.8 |

