# OpenReview forum: "V2P-Bench: Evaluating Video-Language Understanding with Visual Prompts for Better Human-Model Interaction"
_ICLR.cc/2026/Conference — ICLR 2026 Poster_

### Official Review · Reviewer_VvkZ · 2025-10-25

**Soundness:** 2
**Presentation:** 3
**Contribution:** 2
**Rating:** 2
**Confidence:** 3

**Summary:**

The paper introduces a new video benchmark that tests models’ abilities at visual question answering while allowing for visual prompting in addition to text.  With visual prompting, users are allowed to annotate the video and ask questions that refer to the annotation (eg., what is the person with the box around them doing?).  The benchmark is built on existing video collections and includes a range of different types of questions, ranging from low level questions about, for example, object attributes, to higher level questions, about plot or causal relationships.  It seems that all questions are designed to require identification of specific people or objects that might be difficult to otherwise describe.  A large number of open and closed models are identified on the benchmark, which still demonstrate performance that is below human level.

**Strengths:**

The evaluations seem thorough.  Visual prompting does not seem to have been studied adequately in video.  The work seems very comprehensive.

**Weaknesses:**

First, it is not clear how important this problem is.  The questions in the dataset seem designed to benefit from visual prompting.  But how often do users really want to ask questions about people or objects that are difficult to describe?  This might happen often, but there is no evidence given that this is the case.

Second, it is not clear that the results that visual prompting is more effective and efficient than text prompting are truly valid, since the questions developed seem designed to favor visual prompting.  This would be more convincing if these questions arose organically from users attempting to solve real tasks.

Third, the fact that the benchmark contains high and low level questions does not seem very significant.  Visual prompting is used entirely to make it easier for a user to specify a person or object.  The only real question is how much more effective this is than specifying people or objects with text.  If the model can interpret the visual prompting, failures to answer higher level questions have nothing to do with the prompt, but more to do with higher level reasoning.

Four, one of the key questions in video understanding is the extent to which models can integrate information from different parts of the video.  Video prompting seems to be primarily used here with questions that are temporally localized, and could perhaps be answered using a single frame of the video.

**Questions:**

Please provide more details on how the questions were chosen.  How do you ensure that you are not selecting questions that will bias performance in favor of visual prompting?

Experiments in Table 3 are not clearly explained.  How are the text and visual prompts generated?  In Table 3b, what interface is provided for visual prompts?  Is training required?  The supplementary material addresses some of this, but I still found it unclear.  This should be clearly explained in the body of the paper.

---

> ### Author Response · Authors · 2025-11-21
> **Responses to Official Review by Reviewer VvkZ [1/3]**
>
> Thank you for your insightful feedback. We respond to each issue point-by-point below and supplement our answers with further analysis where needed.
>
> > **W1**: First, it is not clear how important this problem is. The questions in the dataset seem designed to benefit from visual prompting. But how often do users really want to ask questions about people or objects that are difficult to describe? This might happen often, but there is no evidence given that this is the case.
>
>
> Building on our original user study, we updated the interaction procedure and conducted a new round of experiments to further examine **how frequently users naturally ask questions about people or objects that are difficult to describe textually**. Specifically, the human–model interaction workflow consisted of the following steps: (1)participants watched a video, (2)formulated a meta-question, (3)rewrote the question into both a text version and a visual-prompt version, and then (4)completed the QA tasks. Afterward, participants were asked to indicate whether they would **rather use text or visual prompts** to formulate questions. We also recorded **the order** in which they produced the text question and the visual-prompt question.
>
> We recruited 20 new volunteers to participate in this experiment. **The participants were not informed of the purpose of the study so there is no bias in theory.** They were only responsible for the interaction tasks so there is no bias in the experiment. All other experimental settings remained consistent with those in the paper. We recorded **five key metrics**:
>  Answer accuracy(maximum 100), Task completion time(seconds), Overall user satisfaction(maximum 10), User preference and Question formulation order.
>
> The experimental results are as follows:
>
> | Metric            | Text Prompt | Visual Prompt |
> |-------------------|-------------|----------------|
> | Acc               | 57.0        | 69.5           |
> | Cost Time         | 25.2        | 18.1           |
> | User Satisfaction | 5.3         | 7.5            |
>
> | User Preference | Text Prompt | Visual Prompt | No Preference |
> |------------------|-------------|----------------|----------------|
> | Nums             | 57          | 129            | 14             |
> | Percentage       | 28.5%       | 64.5%          | 7.0%           |
>
> | Question Order | Text First | Visual First |
> |----------------|------------|---------------|
> | Nums           | 64         | 136           |
> | Percentage     | 34.0%      | 68.0%         |
>
>
> For **Acc**, **Cost Time**, and **User Satisfaction**, the conclusions remain consistent with the first user study: visual prompts provide clear advantages in human–model interaction, improving accuracy, efficiency, and user satisfaction.
>
> Moreover, the updated user study shows that in real interactive scenarios, visual prompts are not a “theoretical concept” but a **genuinely needed interaction modality**. Among the 200 valid question-generation cases, **64.5%** of participants explicitly preferred using visual prompts, while only **28.5%** preferred text-only input, and **7.0%** had no preference.
>
> What's more, **68%** of the questions were initiated using visual prompts. This indicates that for real users, visually selecting the target is more intuitive and less effortful than crafting a natural-language description, especially for objects, locations, or people that are difficult to describe precisely with text.
>
> These findings demonstrate that **visual prompts are actually needed in practice** and offer substantial benefits for both model performance and the overall interaction experience. Users’ unfamiliarity with visual prompts mainly stems from the **lack of sufficiently convenient UIs** to support them at present.
>
> We have incorporated the updated experimental setup, results, and analysis into **Table 3** and **Appendix E.2** of the revised manuscript to further demonstrate that visual prompts are more user-friendly in human–model interaction and are preferred over pure text prompts.

---

> ### Author Response · Authors · 2025-11-21
> **Responses to Official Review by Reviewer VvkZ [2/3]**
>
> > **W2 & Q1**: Second, it is not clear that the results that visual prompting is more effective and efficient than text prompting are truly valid, since the questions developed seem designed to favor visual prompting. This would be more convincing if these questions arose organically from users attempting to solve real tasks.
>
>
> Our benchmark is constructed from 12 publicly available video datasets, all of which were originally **annotated with pure text-based QA pairs**. The annotators of these datasets did not assume that “users will provide visual prompts” when creating the annotations, meaning that the data are **naturally more aligned with text descriptions**. The visual-prompt version of the benchmark is a rewrite we created on top of these original annotations.
> Therefore, from the perspective of data provenance:
>
> ➡ **If any bias exists, it should favor pure text, not visual prompts.**
>
> Furthermore, to ensure a fair comparison between text prompts and visual prompts, we uniformly rewrote all text prompts rather than directly reusing the original textual annotations. Since the 12 constituent datasets vary widely in linguistic style, granularity, and descriptive conventions, directly comparing visual prompts with the original text would introduce additional bias. To avoid this, we rewrote all questions into a standardized text-prompt format and ensured **the text prompts contained the same information as the visual prompts**.
>
> Under the condition, where any bias should logically favor text prompts, experiments still show that converting text prompts into visual prompts **significantly improves accuracy across all models**, with the largest improvement of **+15.1%** observed on Gemini-2.5-Pro, showing that **The advantage of visual prompts does not come from the dataset itself, but from their more natural and efficient ability to specify targets in complex scenes.**
>
>
> On the other hand, to verify the practical value of visual prompts in real interactive scenarios, we conducted a user experience study (Appendix C.2). Since **all user-generated questions originate naturally from real task needs** rather than being artificially constructed, the study possesses strong authenticity and credibility. The results show that under this setting, visual prompts improve model performance by **+13.5%** compared to text-only prompts, and also significantly enhance interaction efficiency and overall user experience. This demonstrates that **visual prompts are both model-friendly and user-friendly**. The updated experiment referenced in **W1** is consistent with the conclusions of the initial study.
>
>
> |                    | Text Prompt | Visual Prompt |
> |--------------------|-------------|----------------|
> | **Acc**            | 62.0        | 75.5           |
> | **Cost Time**      | 27.4        | 19.7           |
> | **User Satisfaction** | 5.8      | 8.2            |
>
> In the revised manuscript, we provide a detailed discussion of the potential biases in the dataset in **Appendix A.6**.
>
> > **W3**: Third, the fact that the benchmark contains high and low level questions does not seem very significant. Visual prompting is used entirely to make it easier for a user to specify a person or object. The only real question is how much more effective this is than specifying people or objects with text. If the model can interpret the visual prompting, failures to answer higher level questions have nothing to do with the prompt, but more to do with higher level reasoning.
>
> Our benchmark is not merely designed to check whether a model can “see” a visual prompt, nor is it limited to low-level visual matching tasks. Instead, it is built as a **comprehensive diagnostic framework** for multimodal models. The multi-level, multi-dimensional task hierarchy is not decorative. It is essential for ensuring that the benchmark remains robust, diagnostically meaningful, and aligned with real human–model interaction scenarios, even as models continue to evolve.
>
> In real-world interactions, user questions rarely stop at basic attribute recognition; they naturally progress toward temporal understanding, causal reasoning, and other higher-level cognitive demands. Therefore, it is crucial to design a benchmark that covers **both low-level perceptual skills and high-level reasoning abilities**, enabling us to distinguish whether a model fails because it “cannot perceive” or because it “cannot reason.”
>
> Furthermore, in **Section 4.3 Error Analysis**, we systematically decompose model failures in visual prompt understanding into Perception Errors, Reasoning Errors, and Hack Errors.
>
> The contribution of V2P-Bench goes far **beyond “making it easier for models to identify user-specified targets.”** It serves to quantify whether visual prompts genuinely improve model understanding and guide future improvements in visual representations, temporal modeling, instruction following, and reasoning.

---

> ### Author Response · Authors · 2025-11-21
> **Responses to Official Review by Reviewer VvkZ [3/3]**
>
> > **W4**: Four, one of the key questions in video understanding is the extent to which models can integrate information from different parts of the video.
>
>
> Beyond the need to **integrate information across different parts of a video**, video understanding also involves **needle-in-a-haystack retrieval tasks**, where a model must accurately identify a key moment or target hidden within large amounts of redundant, noisy, or irrelevant frames. Such tasks require the model to perform efficient and robust target localization within broad semantic contexts. Notably, the Gemini technical report`[1]` also introduces a benchmark specifically designed to evaluate a model’s *needle-in-a-haystack capability*, underscoring the importance of this challenge.
>
> As shown in Section 3.1, **BP tasks** are classic needle-in-a-haystack retrieval tasks that can be solved using only a single frame, whereas **TU and HR tasks** genuinely require the model to integrate information across multiple temporal segments of a video. We design these tasks to simultaneously evaluate two key capabilities of the model: its ability to integrate information across different parts of a video, and its needle-in-a-haystack retrieval ability.
>
> > **Q2**: How are the text and visual prompts generated? In Table 3b, what interface is provided for visual prompts? Is training required?
>
> * **Visual Prompt Annotation**:
>
> To enable the annotation of visual prompts, we implemented a lightweight labeling system based on **Gradio**. The system’s UI supports displaying, playing, and pausing the video at the top. When the video is paused, clicking again enlarges the current frame and automatically launches **MS Paint**, allowing annotators to draw visual prompts directly on the video frame. The UI also displays the original QA pair and allows annotators to type in new questions when necessary.
>
> To better approximate real-world distributions, **all visual prompts were annotated fully manually**, rather than using any predefined or standardized shapes.
>
> * **Text Prompt Annotation**:
>
> After completing the annotation of the visual-prompt dataset, we **uniformly rewrote all text prompts**, rather than directly reusing the original textual annotations. The twelve source datasets differ significantly in linguistic style, granularity, and description patterns; therefore, directly comparing their original questions would introduce additional bias. To ensure fairness, we rewrote all questions into a standardized text-prompt format and guaranteed that **the amount of information conveyed matched that of the visual-prompt version.**
>
> Concretely, we converted the visual-prompt references in each QA pair into **natural-language descriptions of the target’s appearance**. For example, a question such as *“What does the **arrow-pointed person** do after getting off the car?”* was rewritten as *“What does **the person wearing a black suit and black hat** do after getting off the car?”* In contrast, in the original text-only dataset, the same question appeared as *“What does the **main character** in the video do after getting off the car?”*—a formulation that requires the model to first understand the video content and identify the main character. This would be unfair when comparing text prompts to visual prompts, since the latter already explicitly localize the target.
>
> Therefore, we performed a unified rewriting of all text prompts to eliminate this bias and ensure a fair comparison between text-based and visual-prompt–based evaluation.
>
> * **User Experience Study**:
>
> In fact, we **reused Gradio annotation system** that was employed during dataset construction, and the volunteers interacted with the exact same interface as the annotators. The system UI allows users to display, play, and pause the video at the top. Clicking again on a paused frame enlarges the frame and opens MS Paint, enabling users to draw visual prompts directly on the video. The UI also displays the original QA pair and allows participants to type new questions. After the user submits a query, the video, the visual-prompt frame, and the question are sent to the Gemini-2.5-Pro API to obtain a model response. We made slight improvements to the system to **support displaying the model’s response** directly within the interface.
>
> **To avoid potential bias or preference** that professional annotators might have due to their familiarity with the dataset construction process, we recruited volunteers to participate in the user study. Before beginning the experiment, **each volunteer received instructions** from an annotator and completed one practice interaction under supervision. Only after this training step did the volunteers proceed to the actual study.
>
> We have incorporated the revised version of the user study into **Section 4.2 and Appendix E.2** of the updated manuscript to provide additional experimental details.
>
> `[1]` Gemini 1.5: Unlocking multimodal understanding across millions of tokens of context

---

> > ### Comment · Reviewer_VvkZ · 2025-11-26
> >
> > The authors have provided a comprehensive response to Q1 with experiments that are convincing.  There response to Q2 is also convincing.  I do not find their responses to Q3 and Q4 completely persuasive, but I do not think that these issues are as critical.  I have raised my score to 6.

---

> > > ### Author Response · Authors · 2025-11-27
> > > **Responses to Official Review by Reviewer VvkZ [1/2]**
> > >
> > > Thank you very much for the time and effort you invested in the review process, and for your detailed feedback and positive evaluation of our responses to Q1 and Q2. For Q3 and Q4, we provide further analysis and clarification below.
> > >
> > > > W3: Third, the fact that the benchmark contains high and low level questions does not seem very significant. Visual prompting is used entirely to make it easier for a user to specify a person or object. The only real question is how much more effective this is than specifying people or objects with text. If the model can interpret the visual prompting, failures to answer higher level questions have nothing to do with the prompt, but more to do with higher level reasoning.
> > >
> > >
> > > We categorize our dataset into three types of tasks: **Basic Perception (BP)**, **Temporal Understanding (TU)**, and **High-level Reasoning (HR)**. Beyond evaluating whether a model can correctly interpret visual prompts (BP), we also aim to assess its temporal reasoning ability (TU) and high-level reasoning capability (HR) within visually prompted tasks. These tasks form an inherent cascade: the model must first perceive the visual prompt accurately before it can perform temporal integration and logical inference.
> > >
> > > **This design reflects real-world human–model interaction**, where user queries rarely stop at basic attribute recognition. Instead, they naturally evolve toward temporal understanding, causal reasoning, and other higher-level cognitive demands. Therefore, a comprehensive evaluation across these task levels is both necessary and essential.
> > >
> > > To address the question of how much more effective visual prompting is compared to specifying a person or object through text, we reorganize the results from Table 3(a) by task type, as shown below:
> > >
> > > | Model         | **Text** |   **Prompt**   |      |      | **Visual** |   **Prompt**   |      |      |
> > > |---------------|------------------|------|------|------|-------------------|------|------|------|
> > > |               | BP   | TU   | HR   | Avg  | BP   | TU   | HR   | Avg  |
> > > | Qwen2.5-VL-7B | 49.3 | 36.5 | 38.2 | 43.1 | 58.3 | 45.8 | 47.4 | 52.4 |
> > > | Mimo-VL-7B    | 52.2 | 39.6 | 42.8 | 46.7 | 61.4 | 48.3 | 51.9 | 55.6 |
> > >
> > > **BP tasks directly reveal the effectiveness of visual prompts** compared with textual specification of a person or object, since they focus solely on understanding the target indicated by the prompt. **TU and HR tasks can likewise fairly reflect the advantage of visual prompting**. The only difference between the two settings remains the type of prompt used (text vs. visual).
> > >
> > > Note that errors caused by insufficient high-level capabilities only affect the upper bound of a model’s performance on these tasks. In other words, for a subset of samples, the model may fail even if it correctly understands the prompt, simply because it lacks the required temporal or reasoning abilities. However, once we exclude these samples, the remaining data points are those for which the model already possesses the necessary skills to solve the task. In these cases, whether the model answers correctly depends solely on whether it has properly understood the prompt, **allowing for a fair comparison of the effectiveness of different prompt types**.

---

> > > > ### Author Response · Authors · 2025-11-27
> > > > **Responses to Official Review by Reviewer VvkZ [2/2]**
> > > >
> > > > > W4: Four, one of the key questions in video understanding is the extent to which models can integrate information from different parts of the video.
> > > >
> > > > We fully agree that one of the core challenges in video understanding lies in whether a model can effectively integrate key information from different segments of a video. At the same time, **“needle-in-a-haystack” retrieval task** is equally crucial. The model must accurately pinpoint critical moments or targets that are buried within large amounts of redundant, noisy, or irrelevant frames. Notably, the Gemini technical report[1] also introduces a benchmark specifically aimed at evaluating a model’s needle-in-a-haystack capability, further underscoring the significance of this challenge.
> > > >
> > > > To quantitatively evaluate both types of tasks on our dataset, we follow EgoSchema`[1]` and compute the **temporal certificate length** and the **temporal certificate ratio** (the proportion of the minimal effective content duration relative to the total video duration) based on the video length and task type in our dataset. The results are shown in the table below.
> > > >
> > > > | |**Short**|  |  |**Medium**  |  |  |**Long** |  |  |
> > > > |----------------------|----------------------------------|------------------------|------------------------|----------------------------------|------------------------|------------------------|----------------------------------|------------------------|------------------------|
> > > > | | BP | TU| HR| BP | TU| HR| BP | TU| HR|
> > > > | **Certificate Length**| 1s | 31s  | 27s  | 1s | 204s | 169s | 1s | 1147s  | 984s |
> > > > | **Certificate Length Percentage** | 1.9%  | 61.4%| 52.9%| 0.2%  | 44.7%| 37.2%| 0.003%| 35.3%| 30.3%|
> > > >
> > > >
> > > > **Basic Perception(BP)** is designed solely to assess a model’s fundamental ability to perceive visual prompts and can therefore be completed using only a single prompted frame; it falls under the category of **needle-in-a-haystack tasks**. In contrast, **Temporal Understanding(TU)** and **High-level Reasoning(HR)** focus on dynamic information and chronological relationships in videos, as well as on logical inference and higher-level judgment. These tasks require multiple video frames to be properly resolved and therefore fall into the category of **information-integration tasks**.
> > > >
> > > > We evaluate Qwen2.5-VL-7B across **different task types and video durations** to investigate how varying sampling frame rates affect model performance. Here, 1 frame indicates that only the visual-prompt frame is provided to the model. The results are shown in the table below:
> > > >
> > > > | Model| Frames | Short  | | |  Medium  | | |  Long| | |
> > > > |---------------|--------|---------------------|-------|-------|------------------------|-------|-------|------------------------|-------|-------|
> > > > ||  | BP| TU | HR | BP| TU | HR | BP| TU | HR |
> > > > | Qwen2.5-VL-7B | 1| 62.1 | 28.6  | 29.8  | 64.3 | 30.4  | 31.1  | 47.6 | 27.4  | 33.2  |
> > > > || 8| 61.8 | 38.3  | 43.0  | 59.1 | 43.6  | 44.2  | 44.8 | 41.7  | 45.7  |
> > > > || 64  | 60.8 | 40.1  | 46.8  | 56.1 | 47.9  | 51.8  | 43.1 | 41.7  | 50.0  |
> > > >
> > > >
> > > > We observe that for the **BP task**, which falls under the category of needle-in-a-haystack tasks, **increasing the sampling frame rate actually leads to a certain degree of performance degradation**. This is because more frames enlarge the temporal search space, making it harder for the model to accurately locate the visual prompt frame. In contrast, **TU and HR tasks** fall into the category of information-integration tasks. As the frame rate increases, the model gains access to richer dynamic cues, resulting in **a clear performance improvement** that eventually saturates.
> > > >
> > > > We are wondering if the responses have addressed your concerns and we would greatly appreciate it if you could consider them in your final rating. We look forward to your further feedback.
> > > >
> > > >
> > > > `[1]` EgoSchema: A Diagnostic Benchmark for Very Long-form Video Language Understanding

---

### Official Review · Reviewer_8395 · 2025-10-29

**Soundness:** 3
**Presentation:** 3
**Contribution:** 3
**Rating:** 6
**Confidence:** 4

**Summary:**

This paper proposes V2P-Bench, a new benchmark addressing the inefficiency of text-based evaluation in video-language understanding. By using visual prompts directly on video frames, it assesses model performance across perception, temporal, and reasoning tasks. Results show that visual prompts improve model accuracy while exposing weaknesses in long video and spatiotemporal reasoning.

**Strengths:**

This paper introduces the novel idea of using visual prompts for video-language evaluation, providing a more intuitive and human-aligned alternative to text-based prompts. The dataset and tasks are well designed, with careful annotation and broad experimental coverage across major LVLMs. The presentation is clear and well-structured, supported by effective figures and analyses.

**Weaknesses:**

The main weaknesses of this paper lie in its theoretical and methodological depth. While the idea of visual prompts is novel, the work lacks a stronger theoretical or cognitive grounding to explain why this approach better reflects human interaction. The evaluation remains limited to offline video QA with a relatively small dataset and does not include audio or multimodal signals, which limits its realism and scalability. In addition, the analysis is mostly descriptive, without deeper investigation into why models fail in spatiotemporal reasoning or specific error types.

**Questions:**

I have several critical questions that the authors should address to clarify the scientific validity of the work.

First, the central claim, that visual prompts are a more human-aligned form of interaction, remains unsubstantiated. Could the authors provide theoretical grounding or empirical evidence (e.g., from cognitive studies or user experiments) rather than relying on intuitive justification?

Second, the benchmark design is still based on offline video QA, which seems inconsistent with the paper’s framing around “human–model interaction.” How do the authors ensure that this static setup meaningfully reflects real interactive understanding?

Third, the error analysis is largely descriptive and lacks mechanism-level insight. Why do models consistently fail on spatiotemporal reasoning tasks, and how can the authors be sure that the evaluation protocol effectively eliminates “hack” behaviors rather than concealing them?

Addressing these questions is essential to assessing the scientific rigor and actual contribution of the work.

---

> ### Author Response · Authors · 2025-11-21
> **Responses to Official Review by Reviewer 8395 [1/3]**
>
> We sincerely appreciate your insightful feedback and thoughtful suggestions. Below, we respond to each concern in detail and provide additional analysis and clarification.
>
> > **Q1**: First, the central claim, that visual prompts are a more human-aligned form of interaction, remains unsubstantiated. Could the authors provide theoretical grounding or empirical evidence (e.g., from cognitive studies or user experiments) rather than relying on intuitive justification?
>
>
> In Section 4.2, we raise a key question: **“Which prompt type works better for humans and models?”** To answer this, we designed two experiments and provided corresponding quantitative results and analyses. For ease of review, we summarize the relevant content below:
>
> **(1) Visual prompts are more model-friendly than text prompts.**
>
> Based on our constructed visual-prompt dataset, we *rewrote* the visual-prompt descriptions in each QA pair into natural-language descriptions of the target’s appearance, thereby creating a text-only version of V2P-Bench. The detailed experimental setup is provided in Appendix C.2. We then evaluated the model’s performance under text-only prompts versus visual prompts, as shown in the table below:
>
> | Model| Text Prompt | Visual Prompt |
> |----------------|-------------|----------------|
> | GPT-4o    | 53.0   | **65.4** |
> | Gemini-2.5-Pro | 54.7   | **69.8** |
> | LLaVA-Video-7B | 42.4   | **54.8** |
> | Qwen2.5-VL-7B  | 43.1   | **52.4** |
> | Mimo-VL-7B| 46.7   | **55.6** |
>
> We observe that **simply converting visual prompts into text prompts leads to a substantial drop in accuracy** for all evaluated models, with the most significant decrease of **15.1%** occurring in Gemini-2.5-Pro. This degradation arises because text prompts require the model to decode the target solely from linguistic descriptions, which increases the difficulty of correctly interpreting the user’s intent. In contrast, visual prompts can directly and precisely localize the target within the video frames, eliminating both the user’s need to encode their intent in text and the model’s need to infer it through language.
>
> **(2) Visual prompts are more user-friendly than text prompts.**
>
> To assess the practical usability of visual prompts, we conducted a **User Experience Study**. The detailed experimental setup can be found in Appendix C.2, and the results are presented in Table 3(b):
>
> | Metric| Text Prompt | Visual Prompt |
> |-------------------|-------------|----------------|
> | Acc| 62.0  | **75.5**  |
> | Cost Time| 27.4  | **19.7** |
> | User Satisfaction | 5.8| **8.2**|
>
> As shown, users complete tasks more quickly in the visual prompt interaction setting, with the time reduced from 27.4s to 19.7s, saving **7.7s** on average. In addition, the improvement in model performance means more satisfactory responses for users. Together, these factors contributed to an average user satisfaction score of 8.2, **2.4 points** higher than under the text-only prompt condition. This indicates that **visual prompts provide clear advantages in human–model interaction, improving both overall satisfaction and efficiency**.

---

> > ### Author Response · Authors · 2025-11-21
> > **Responses to Official Review by Reviewer 8395 [2/3]**
> >
> > > **Q2**: Second, the benchmark design is still based on offline video QA, which seems inconsistent with the paper’s framing around “human–model interaction.” How do the authors ensure that this static setup meaningfully reflects real interactive understanding?
> >
> >
> > Thank you for raising this important concern. In the Limitation Section of Appendix F, we discuss the inherent limitations of our current benchmark. We note that “it only focuses on visual and textual inputs, lacks audio input, and supports evaluations only on offline videos, leaving a gap compared to the ultimate form of multimodal human–model interaction in the real world.” A fully interactive multimodal system we think should include: Comprehensive video understanding, Full-modality integration, Multi-turn conversational interaction and Interruptible and real-time interaction(like VITA).
> >
> >
> > As the first benchmark to incorporate **visual prompts** into human–model interaction for video understanding, V2P-Bench addresses key limitations of existing text-only evaluation frameworks. Under the **current offline-video evaluation** paradigm, we arrive at the following findings:
> >
> > * **Zero-shot visual prompt understanding is far from solved.**
> >   Even the strongest proprietary models perform significantly below human experts, indicating substantial room for future progress within the community.
> >
> > * **Visual prompts outperform text prompts.**
> >   Through our experiments comparing different prompt types and our user experience study (Q1), we demonstrate that **visual prompts are both more model-friendly and more user-friendly** than text prompts.
> >
> > * **Error diagnosis becomes feasible.**
> >   We find that **57.7%** of failures arise from pure perception errors rather than dialog flaws, offering actionable guidance for improving future interactive multimodal systems.
> >
> >
> > In **Appendix E**, we also discuss practical use cases of visual prompts under the current offline evaluation framework:
> >
> > * **Interaction with Mobile Devices.**
> > Users can watch videos on mobile devices such as smartphones and computers, and easily create visual prompts using their fingers or a mouse. This functionality has already been implemented in systems like GPT-4o`[1]` and Video Refer`[2]`, enabling users to interact with video content more intuitively and enhancing their overall viewing experience.
> >
> > * **Smart Device Wearables.**
> > Imagine users wearing advanced smart devices (e.g., Apple Vision Pro) and being immersed in a vast stream of visual content. Such users could generate visual prompts through natural gestures. This interaction paradigm opens up immense possibilities for the future, greatly enriching the immersive video-watching experience.
> >
> > The current benchmark serves as a diagnostic pioneer, quantifying the level of visual-prompt understanding required by future interactive systems. Even under a static evaluation setting, it already brings tangible improvements to the user experience. Looking forward, we plan to develop V2P-Bench v2, which will support all types of video understanding, incorporate full-modality inputs, and enable the evaluation of multi-turn dialogue and interruptible interactions.

---

> > > ### Author Response · Authors · 2025-11-21
> > > **Responses to Official Review by Reviewer 8395 [3/3]**
> > >
> > > > **Q3**: Third, the error analysis is largely descriptive and lacks mechanism-level insight. Why do models consistently fail on spatiotemporal reasoning tasks, and how can the authors be sure that the evaluation protocol effectively eliminates “hack” behaviors rather than concealing them?
> > >
> > > More than half of the errors (57.7%) stem from failures in visual perception. These issues may relate to insufficient coverage in the model’s training data as well as limited capability in maintaining cross-frame object consistency, both of which directly undermine the model’s performance on higher-level temporal reasoning tasks.
> > >
> > >
> > >
> > > In the V2P-Bench evaluation pipeline, we introduce an additional diagnostic experiment in which the video–question pairs are randomly shuffled. This setup ensures that the model can no longer rely on the video content to derive the correct answer; therefore, an honest model should refuse to answer all questions. We conduct this test using Qwen2.5-VL-7B and MiMo-VL-7B. **The Trigger Ratio denotes the proportion of cases in which the model chooses to refuse answering.** We perform standard inference and count all responses that do not select any option. The results are shown below:
> > >
> > > | Model           | Trigger Ratio |
> > > |-----------------|----------------|
> > > | Qwen2.5-VL-7B   | 6.4%           |
> > > | MiMo-VL-7B      | 3.9%           |
> > >
> > > Under this setting, the models should, in principle, refuse to answer all questions. However, both models still provide answers in more than **90%** of the cases, indicating that they forcibly choose an option even without valid visual context. This offers direct evidence of the presence of Hack behavior in the models and **fully reveals such behavior rather than concealing it**.
> > >
> > >
> > > `[1]` GPT-4o system card
> > >
> > > `[2]` VideoRefer Suite: Advancing Spatial-Temporal Object Understanding with Video LLM

---

> > > > ### Comment · Reviewer_8395 · 2025-11-26
> > > >
> > > > Thanks for the clear replies. I read through your explanations and also checked the discussions with the other reviewers. I think you addressed the main points well. I’ll keep my original score (6, borderline accept).
> > > >
> > > > One small comment: it seems iclr allows extending the main paper to 10 pages during rebuttal, so some key clarifications could also be moved into the main text. I noticed most updates are currently in the appendix, not a big issue, just a suggestion.

---

> > > > > ### Author Response · Authors · 2025-11-26
> > > > >
> > > > > Thank you very much for your response and we are delighted to receive your feedback!  We have updated and expanded the main text and the appendix based on the rebuttal, which are now available for review at your convenience.
> > > > >
> > > > > We would be very happy to provide further clarification if you have any other questions.
> > > > >
> > > > > Best regards,
> > > > >
> > > > > Paper 8969 authors

---

### Official Review · Reviewer_9ULh · 2025-10-31

**Soundness:** 3
**Presentation:** 3
**Contribution:** 3
**Rating:** 6
**Confidence:** 4

**Summary:**

The paper proposes V2P-Bench, a robust and comprehensive benchmark for evaluating the ability of LVLMs to understand video visual prompts in human–model interaction scenarios. The authors argue that existing video benchmarks primarily rely on text prompts, which often involve complex referential language and, in turn, reduce the accuracy and efficiency of human–model interaction. To address this, they introduce more user-friendly visual prompts. The authors conduct a comprehensive analysis and highlight several key findings, such as performance on spatiotemporal understanding and the prevalence of the hack phenomenon.

**Strengths:**

1. The paper is well-motivated and studies a timely problem of visual prompt-based evaluation instead of unnecessarily overcomplicated textual prompts. I particularly liked that the authors clearly established this with a human study as well.
2. The benchmark is carefully curated with multiple automated and human checks.
3. The authors conduct comprehensive experiments to study different models, and go beyond standard evaluation to study interesting hack phenomenon and other ablations.

**Weaknesses:**

While I don't foresee any major weaknesses, I have some questions about experiment design and further experiments that I would like to see before I raise my score further.

**Questions:**

1. The hack phenomenon is concerning, especially as it increases rates as the video length grows. I am curious if the authors convert their benchmark to an open-ended generation one instead of MCQ-based, would that help in reducing some of this?

2. I am curious how does the structuring of the visual query itself impact the performance? For instance, what if you have scribbled unstructured bounding boxes instead of perfect rectangles or other more natural markers that are more user-friendly, how does that impact overall performance?

3. From many of the query examples provided in the paper, it appears that most could be resolved using only a few frames, with the main bottleneck being the grounding of the visual query frame. Could the authors elaborate on this point, perhaps by quantifying it using temporal certificates [1] or frame rate ablations? Specifically, do the models on their benchmark perform better as more visual information is provided, or can the benchmark generally be answered using just a few frames?

[1] EgoSchema: A Diagnostic Benchmark for Very Long-form Video Language Understanding

---

> ### Author Response · Authors · 2025-11-21
> **Responses to Official Review by Reviewer 9ULh [1/3]**
>
> We sincerely thank you for your constructive feedback and valuable suggestions. Below we address the raised concerns point-by-point and provide additional analyses and clarifications.
>
> > **W1**: The hack phenomenon is concerning, especially as it increases rates as the video length grows. I am curious if the authors convert their benchmark to an open-ended generation one instead of MCQ-based, would that help in reducing some of this?
>
>
>
> Existing benchmarks commonly adopt the MCQ format because it is **easy to evaluate and highly controllable**, and avoids the ambiguity often seen in open-ended responses. Our experimental results show that models indeed exhibit such behavior: **they tend to game the MCQ structure to obtain higher scores instead of performing true video comprehension**. This finding suggests that, in addition to pursuing higher benchmark scores, future models should also prioritize genuine understanding and groundedness as key optimization objectives.
>
> To examine **whether open-ended generation helps reduce hack behaviors**, we conducted two experiments in which we removed the multiple-choice constraints and answer options from the original prompts, converting them into open-ended question–answering tasks:
>
>
> **(1) Random Shuffling Experiment**
>
> We randomly shuffle the videos and questions, meaning the model can no longer obtain the answer from the video content. If a model is sufficiently honest, it should refuse to answer all questions. We perform inference using Qwen2.5-VL-7B and MiMo-VL-7B. **The Trigger Ratio denotes the proportion of cases in which the model refuses to answer**. For the MCQ setting, we perform standard reasoning and count all responses that do not select any option; for the OE setting, we manually evaluate whether the model refused to respond. The results are shown in the table below:
>
> | Model |MCQ  | Trigger |  Ratio|| OE|  Trigger|Ratio  ||
> |---------------|----------------------------------|--------------|-------------|-------|-----------------------------------|--------------|-------------|-------|
> || Short| Medium| Long| Avg| Short| Medium| Long| Avg|
> | Qwen2.5-VL-7B | 6.2% | 6.9% | 6.3%| 6.4%  | 95.8% | 96.8%| 96.1%| 96.7% |
> | MiMo-VL-7B| 3.6% | 4.2% | 4.0%| 3.9%  | 92.2% | 94.6%| 94.8%| 93.6% |
>
>
> We observe that under the MCQ setting, Qwen2.5-VL-7B and MiMo-VL-7B exhibit refusal rates of only **6.4%** and **3.9%**, respectively. This indicates that even when the video and question are completely mismatched, the models still select an option in the vast majority of cases, and the trigger rates remain almost identical across different video lengths. In contrast, under the OE setting, the refusal rates rise sharply to **96.7%** and **93.6%**, again showing little variation across video lengths. These results clearly demonstrate that **open-ended generation effectively mitigates hack behaviors to a large extent, fully confirming your hypothesis**.
>
>
> **(2) Model Performance on V2P-Bench**
>
> Besides，we follow the same setup as in the random shuffling experiment, with the only difference being that we use the original benchmark data rather than the shuffled version. The experimental results are shown in the table below:
>
> | Model |MCQ|  Hack|  Ratio|| OE| Hack | Ratio ||
> |---------------|-------------------------------|--------------|--------------|-------|---------------------------------|--------------|--------------|-------|
> || Short | Medium| Long | Avg| Short| Medium| Long | Avg|
> | Qwen2.5-VL-7B | 8.1%  | 11.5%| 14.4%| 10.2% | 2.1%| 2.6% | 3.4% | 2.7%  |
> | MiMo-VL-7B| 5.3%  | 7.2% | 7.4% | 6.2%  | 1.7%| 2.3% | 2.8% | 2.2%  |
>
> We observe that after converting the MCQ task into an open-ended question–answering task, the Hack Ratios of Qwen2.5-VL-7B and MiMo-VL-7B decrease by **7.5%** and **4.0%**, respectively. This further demonstrates that **open-ended generation can substantially mitigate hack behaviors**.
>
> In **Appendix C.2** of the revised manuscript, we provide a detailed analysis and discussion of the hack phenomenon in open-ended evaluation.

---

> ### Author Response · Authors · 2025-11-21
> **Responses to Official Review by Reviewer 9ULh [2/3]**
>
> > **W2**: I am curious how does the structuring of the visual query itself impact the performance? For instance, what if you have scribbled unstructured bounding boxes instead of perfect rectangles or other more natural markers that are more user-friendly, how does that impact overall performance?
>
>
> To investigate **the effect of the visual query structure itself** on the performance of visual prompts, we randomly sampled 217 data instances. For each question–answer pair, we annotated the corresponding visual prompt frames with multiple types of visual prompts, while strictly keeping all other settings identical, including the number of frames, prompts. The results are as follows:
>
> | Method|  Standard  | Shapes |  || Doodle  | Shapes |  ||
> |---------------|------------------------------------|--------------|--------------|-------|----------------------------------|--------------|--------------|-------|
> || Rectangle  | Arrow| SoM  | Avg| Rectangle| Arrow| SoM  | Avg|
> | Qwen2.5-VL-7B | 47.3| 43.6 | 45.1 | 45.3  | 46.7 | 42.9 | 44.4 | 44.6  |
> | MiMo-VL-7B| 54.2| 51.2 | 52.7 | 52.7  | 53.6 | 50.3 | 51.9 | 51.9  |
>
>
> For the same visual prompt type, **the doodle-style shapes are slightly weaker than the standard shapes**. When switching from standard shapes to doodle shapes, the performance of Qwen2.5-VL-7B and MiMo-VL-7B **decreases by 0.7% and 0.8%**, respectively. This is reasonable, since most training data are synthetic, whereas hand-drawn doodles often have unstable boundaries, making it harder for the model to extract consistent visual cues.
>
> Regardless of whether standard or doodle shapes are used, the overall performance trend remains **Rectangle > SoM > Arrow**. Among these, rectangles appear most frequently in training data and can fully enclose the target region, providing a stable and explicit spatial localization signal. In contrast, arrows are typically smaller and carry lower information density, making it more difficult for the model to capture key spatial relationships and therefore resulting in the weakest performance.
>
> In **Section 4.2** of the revised manuscript, we present a detailed experimental analysis of the impact of visual prompt structures.

---

> ### Author Response · Authors · 2025-11-21
> **Responses to Official Review by Reviewer 9ULh [3/3]**
>
> > **W3**: From many of the query examples provided in the paper, it appears that most could be resolved using only a few frames, with the main bottleneck being the grounding of the visual query frame. Could the authors elaborate on this point, perhaps by quantifying it using temporal certificates or frame rate ablations? Specifically, do the models on their benchmark perform better as more visual information is provided, or can the benchmark generally be answered using just a few frames?
>
> Following EgoSchema`[1]`, we compute the **temporal certificate length** and the **temporal certificate ratio** (the proportion of the minimal effective content duration relative to the total video duration) based on the video length and task type in our dataset. The results are shown in the table below.
>
> | |**Short**|  |  |**Medium**  |  |  |**Long** |  |  |
> |----------------------|----------------------------------|------------------------|------------------------|----------------------------------|------------------------|------------------------|----------------------------------|------------------------|------------------------|
> | | BP | TU| HR| BP | TU| HR| BP | TU| HR|
> | **Certificate Length**| 1s | 31s  | 27s  | 1s | 204s | 169s | 1s | 1147s  | 984s |
> | **Certificate Length Percentage** | 1.9%  | 61.4%| 52.9%| 0.2%  | 44.7%| 37.2%| 0.003%| 35.3%| 30.3%|
>
>
> The **Basic Perception** task is designed solely to assess a model’s fundamental ability to perceive visual prompts, and thus it can be completed using a single prompted frame. In contrast, **Temporal Understanding** and **High-level Reasoning** focus on dynamic information and chronological relationships in videos, and on logical inference and higher-level judgment, respectively. As a result, both tasks require multiple video frames to be properly resolved.
>
>
> We evaluate Qwen2.5-VL-7B across **different task types and video durations** to investigate how varying sampling frame rates affect model performance. Here, 1 frame indicates that only the visual-prompt frame is provided to the model. The results are shown in the table below:
>
> | Model| Frames | Short  | | |  Medium  | | |  Long| | |
> |---------------|--------|---------------------|-------|-------|------------------------|-------|-------|------------------------|-------|-------|
> ||  | BP| TU | HR | BP| TU | HR | BP| TU | HR |
> | Qwen2.5-VL-7B | 1| 62.1 | 28.6  | 29.8  | 64.3 | 30.4  | 31.1  | 47.6 | 27.4  | 33.2  |
> || 8| 61.8 | 38.3  | 43.0  | 59.1 | 43.6  | 44.2  | 44.8 | 41.7  | 45.7  |
> || 64  | 60.8 | 40.1  | 46.8  | 56.1 | 47.9  | 51.8  | 43.1 | 41.7  | 50.0  |
>
>
> We observe that for the **BP task**, which relies solely on single-frame information, **increasing the sampling frame rate actually leads to a certain degree of performance degradation**. This is because more frames enlarge the temporal search space, making it harder for the model to accurately locate the visual prompt frame. In contrast, **TU and HR tasks** inherently require multi-frame temporal information. As the frame rate increases, the model gains access to richer dynamic cues, resulting in **a clear performance improvement** that eventually saturates.
>
> In the revised manuscript, **Appendix A.5 and C.3** provide a discussion of the effects of the dataset’s certificate length and the impact of sampling frame rate on model performance.
>
>
> `[1]` EgoSchema: A Diagnostic Benchmark for Very Long-form Video Language Understanding

---

> > ### Author Response · Authors · 2025-11-27
> > **Welcome for Further Feedback**
> >
> > Dear Reviewer 9ULh,
> >
> > Thank you once again for your careful and constructive feedback on our work. During the rebuttal period, we conducted additional analyses and experiments to further address your concerns. The main updates are as follows:
> >
> > * **W1 (Hack behavior on OE task):** We performed additional experiments by converting MCQ tasks into open-ended QA to investigate whether open-ended generation can mitigate hack behaviors. Through the experiment, we found that open-ended generation significantly reduces hack behaviors across models, fully confirming your hypothesis. These results are included in **Appendix C.2**.
> >
> > * **W2 (Structure of visual prompts):** We annotated visual-prompt data with different structures and drawing styles to evaluate and quantify the influence of the prompt structure itself on visual prompting effectiveness. Details are shown in **Section 4.2**.
> >
> > * **W3 (temporal ertificate):** We quantified the temporal certificate of our dataset and conducted frame-rate ablation experiments. The results show that BP relies on single-frame information, and increasing the sampling frame rate can lead to performance degradation. TU and HR benefit substantially from higher frame rates and exhibit clear performance improvements. We provide a detailed discussion in **Appendix A.5 and C.3**.
> >
> > We are wondering if the responses have addressed your concerns and we would greatly appreciate it if you could consider them in your final rating. We look forward to your further feedback.
> >
> > Best regards,
> > ICLR 2026 Paper 8969 Authors

---

### Official Review · Reviewer_rfAK · 2025-10-31

**Soundness:** 2
**Presentation:** 2
**Contribution:** 2
**Rating:** 4
**Confidence:** 4

**Summary:**

The paper introduces V2P-Bench, a benchmark designed to evaluate large vision-language models (LVLMs) using visual prompts instead of text prompts for video understanding, enabling fine-grained assessment of spatial, temporal, and reasoning abilities. Results show that visual prompts improve both model accuracy and user experience, but current LVLMs still lag behind humans, particularly in spatiotemporal comprehension and robustness.

**Strengths:**

- Introduces a visual prompt–based benchmark that improves realism in human–model interaction evaluation.
- Provides a comprehensive dataset with multi-level tasks and rigorous annotation ensuring high quality and diversity.
- Offers clear empirical insights into LVLM weaknesses, including hack phenomena and spatiotemporal reasoning gaps.

**Weaknesses:**

1. The benchmark currently focuses on synthetic QA tasks rather than natural, conversational interactions, which limits its generalization to real-world human–model dialogues. How would you integrate free-form conversational or multimodal dialogue settings to better simulate authentic human–AI interaction? If not, it is really not geberalisable.

2. The dataset construction process is highly manual and time-consuming, making it difficult to scale to larger domains or real-time applications. How can we incorproate scalability?

3. Several analyses, such as the user experience evaluation and hack behavior detection, rely on qualitative interpretation rather than standardized quantitative frameworks. Cou.d the authors design more rigorous, quantitative evaluation protocols and reproducible metrics to strengthen the reliability and comparability of findings?

4. The benchmark primarily evaluates static performance outcomes but does not assess model adaptability or learning progression over time.

**Questions:**

See weaknesses.

**Details Of Ethics Concerns:**

-

---

> ### Author Response · Authors · 2025-11-21
> **Responses to Official Review by Reviewer rfAK [1/3]**
>
> We sincerely appreciate your constructive comments and insightful suggestions. Below, we respond to each concern in detail and provide further analysis and clarification.
>
> > **W1**: The benchmark currently focuses on synthetic QA tasks rather than natural, conversational interactions, which limits its generalization to real-world human–model dialogues. How would you integrate free-form conversational or multimodal dialogue settings to better simulate authentic human–AI interaction? If not, it is really not geberalisable.
>
> We adopt a question–answering format as **QA-based evaluation is easy to automate, highly controllable, and effective at avoiding the ambiguity and vagueness that commonly arise in open-ended responses**. This is also why existing mainstream video-understanding benchmarks(Video-MME`[1]`, MVBench`[2]`, EgoSchema`[3]`, LVBench`[4]`.etc)widely adopt a QA-style design. In contrast, **free-form dialogue introduces substantial subjectivity and is difficult to score in a consistent and reproducible manner**. Moreover, relying on an LLM-as-a-judge may introduce evaluation bias, further weakening the fairness, reliability, and comparability of the assessment.
>
> In addition, to further examine **the consistency between QA tasks and open-ended dialogue**, we remove the multiple-choice constraints from our original questions, convert them into open-ended formats, and re-evaluate the models. The results are shown in the following table:
>
> | Model | MCQ  | OE (GPT-4o Judge) | OE (Human Judge) |
> |----------------|------|--------------------|-------------------|
> | Gemini-2.5-Pro | 69.8 | 48.3| 51.6  |
> | Qwen2.5-VL-7B  | 48.1 | 30.2| 33.7  |
> | MiMo-VL-7B  | 45.7 | 32.7| 35.6  |
>
> As shown, under the open-ended setting, the Pearson consistency coefficient between MCQ and OE (GPT-4o Judge) reaches **0.95**, and the coefficient between MCQ and OE (Human Judge) reaches **0.98**. This demonstrates a strong consistency between QA-style evaluation and free-form dialogue, indicating that **QA-based assessments can reliably reflect a model’s capabilities in open-ended conversational scenarios**.
>
> In **Appendix C.1** of the revised manuscript, we provide a detailed discussion of the consistency of MCQ and dialogue tasks.
>
> > **W2**: The dataset construction process is highly manual and time-consuming, making it difficult to scale to larger domains or real-time applications. How can we incorproate scalability?
>
>
> The core value of a benchmark dataset lies in being **high-quality, trustworthy, and diagnostic**, rather than merely covering as much data as possible. To this end, we rely on trained human annotators instead of automated generation, ensuring that each question maintains a strict causal linkage between the video evidence, visual prompts, and the correct answer. Thus, the benchmark does not prioritize “scaling up data volume” as its primary goal.
>
> On the other hand, **to enable the construction of large-scale visual-prompt datasets and to enhance existing models’ ability** to understand visual prompts, we also provide a fully automated data-generation pipeline, including:
>
> 1. Using **RAM++**`[5]` to automatically extract potential target objects from video;
> 2. Applying **SAM3**`[6]` for cross-frame object tracking and mask propagation to obtain temporally consistent visual annotations;
> 3. Following automatic data-synthesis strategies from **LLaVA-Video**`[7]` and **ShareGPT4Video**`[8]` to bind model-generated QA pairs to target regions, thus producing large quantities of high-quality video–visual-prompt supervision data.
>
> In **Appendix B** of the revised manuscript, we discuss the scalability of the dataset and present feasible solutions.

---

> > ### Author Response · Authors · 2025-11-21
> > **Responses to Official Review by Reviewer rfAK [2/3]**
> >
> > > **W3**: Several analyses, such as the user experience evaluation and hack behavior detection, rely on qualitative interpretation rather than standardized quantitative frameworks. Cou.d the authors design more rigorous, quantitative evaluation protocols and reproducible metrics to strengthen the reliability and comparability of findings?
> >
> > In fact, our paper already includes a **user experience evaluation** (*P7 Table 3*) and **hack-behavior detection experiments** (*P8 Figure 5, P24 Figure 13*), accompanied by corresponding quantitative results and analyses. For clarity, we restate the relevant parts below:
> >
> > **(1) User Experience Study:**
> >
> > To assess the practical usability of visual prompts, we conducted a **User Experience Study**. The detailed experimental setup can be found in Appendix C.2, and the results are presented in *Table 3(b)*:
> >
> > | Metric| Text Prompt | Visual Prompt |
> > |-------------------|-------------|----------------|
> > | Acc| 62.0  | **75.5**  |
> > | Cost Time| 27.4  | **19.7** |
> > | User Satisfaction | 5.8| **8.2**|
> >
> > As shown, users complete tasks more quickly in the visual prompt interaction setting, with the time reduced from 27.4s to 19.7s, saving **7.7s** on average. In addition, the improvement in model performance means more satisfactory responses for users. Together, these factors contributed to an average user satisfaction score of 8.2, **2.4 points** higher than under the text-only prompt condition. This indicates that **visual prompts provide clear advantages in human–model interaction, improving both overall satisfaction and efficiency**.
> >
> >
> > **(2) Hack-Behavior Detection**
> >
> > In Section 4.2, we first conducted a pilot study on hack behavior. We randomly shuffled the videos and questions and performed inference with Qwen2.5-VL-7B and MiMo-VL-7B. As shown in Table 5 on P9, only **6.4% and 3.9%** of the cases triggered a refusal, respectively, while all other cases still followed the instruction to select an option. This indicates that **current models are trained to behave like “test-takers,” often overlooking even basic factual inconsistencies.**
> >
> > | Model           | Trigger Ratio |
> > |-----------------|----------------|
> > | Qwen2.5-VL-7B   | 6.4%           |
> > | MiMo-VL-7B      | 3.9%           |
> >
> >
> >
> > Besides, in Section 4.2 *“Are vision-language models truly understanding videos, or merely exploiting hacks?”*, we conduct a comprehensive investigation into the hack behaviors exhibited by current models and by existing benchmarks. The corresponding empirical results are presented in *P8 Figure 5 and P24 Figure 13* of the paper. Based on our analysis, we draw the following conclusions:
> >
> > **(1) Hack phenomena are prevalent in video benchmark evaluations.**
> > Both Qwen2.5-VL-7B and MiMo-VL-7B consistently display non-zero Hack Ratios across all evaluation settings, accompanied by varying levels of performance degradation.
> >
> > **(2) Longer videos amplify hack behaviors.**
> > For example, under the 4-frame sampling configuration, the Hack Ratios of Qwen2.5-VL-7B increase from **11.1%** (short videos) to **23.0%** (medium videos) and **33.8%** (long videos). This trend remains consistent across other experimental setups.
> >
> > **(3) Fewer sampled frames intensify hack behaviors.**
> > When reducing the sampled frames for Qwen2.5-VL-7B from 128 to 4, the average Hack Ratio steadily rises from **8.0%** to **18.7%**, indicating a monotonic increase as temporal information diminishes.
> >
> > We attribute the emergence of hack phenomena to three factors:
> >
> > **(1) Insufficient temporal information:**
> > Sparse sampling strategies fail to provide the model with enough video evidence to produce grounded answers.
> >
> > **(2) Limited perceptual capability:**
> > Excessive or noisy visual context may obscure the key information, preventing the model from correctly identifying the relevant cues.
> >
> > **(3) Instruction-following–oriented training:**
> > Current models are optimized to comply with instructions, causing them to prioritize selecting an answer—even when the video content is inconsistent—rather than grounding their responses in factual evidence.

---

> > > ### Author Response · Authors · 2025-11-21
> > > **Responses to Official Review by Reviewer rfAK [3/3]**
> > >
> > > > **W4**: The benchmark primarily evaluates static performance outcomes but does not assess model adaptability or learning progression over time.
> > >
> > > Existing VLM benchmarks, including image-based VQA benchmarks (e.g., MMBench, MM-Vet), multimodal mathematical reasoning benchmarks (e.g., MathVerse, MathVista, WeMath), and video benchmarks (e.g., VideoMME, LongVideoBench, EgoSchema),share a common goal to robustly and reproducibly evaluate a model’s static capabilities under fixed conditions.
> > >
> > > From a research perspective, models are responsible for learning and improvement, and benchmarks are responsible for objective evaluation and diagnosis. Enhancing a model’s ability to learn or adapt over time should instead be addressed through data construction strategies, training paradigms, and model architectures. We look forward to future research emerging in areas such as data construction methods, training paradigms, and model architectures, collectively driving continued progress in this field.
> > >
> > >
> > >
> > >
> > > `[1]` Video-MME: The First-Ever Comprehensive Evaluation Benchmark of Multi-modal LLMs in Video Analysis
> > >
> > > `[2]` MVBench: A Comprehensive Multi-modal Video Understanding Benchmark
> > >
> > > `[3]` EgoSchema: A Diagnostic Benchmark for Very Long-form Video Language Understanding
> > >
> > > `[4]` LVBench: An Extreme Long Video Understanding Benchmark
> > >
> > > `[5]` Recognize Anything: A Strong Image Tagging Model
> > >
> > > `[6]` Meta Segment Anything Model 3 (SAM 3)
> > >
> > > `[7]` LLaVA-Video: Video Instruction Tuning With Synthetic Data
> > >
> > > `[8]`ShareGPT4Video: Improving Video Understanding and Generation with Better Captions

---

> > > > ### Author Response · Authors · 2025-11-27
> > > > **Welcome for Further Feedback**
> > > >
> > > > Dear Reviewer rfAK,
> > > >
> > > > We sincerely appreciate your thoughtful and constructive feedback on our submission.  During the rebuttal period, we carefully addressed each of your concerns and conducted additional analyses to strengthen our responses. The key modifications and findings are summarized below:
> > > >
> > > > - **W1 (MCQ vs. free-form dialogue):** We provided empirical evidence showing strong consistency between MCQ-style evaluation and open-ended dialogue (Pearson 0.95/0.98), demonstrating that MCQ-based assessment reliably reflects conversational model capabilities. Details added in **Appendix C.1**.
> > > >
> > > > - **W2 (scalability of dataset):** While maintaining high-quality human-curated benchmark samples, we introduced an end-to-end automated data-generation pipeline that enables scalable construction of large visual-prompt datasets. Extended discussion appears in **Appendix B**.
> > > >
> > > > - **W3 (quantitative evaluation protocols):** We clarified the quantitative user study and hack-behavior detection already included in the paper, and highlighted the systematic findings derived from these experiments. Details are shown in **Section 4.2**.
> > > >
> > > > - **W4 (adaptation and learning over time):** We emphasized alignment with standard practice in existing VLM benchmarks, which focus on static capability evaluation, while model adaptability is addressed through training and data-design paradigms.
> > > >
> > > > We are wondering if the responses have addressed your concerns and would appreciate it if you consider raising your final rating. Looking forward to your further feedbacks!
> > > >
> > > > Best regards,
> > > >
> > > > ICLR 2026 Paper 8969 Author

---

### Author Response · Authors · 2025-12-03
**Summary for AC Consideration**

Dear Area Chair,

We sincerely appreciate your thorough review of our paper and the valuable time and effort you dedicated to the discussion. To help quickly understand our discussion with the reviewers, we summarize the rebuttal phase from two perspectives: the **Reviewers’ Overall Review** and the **Key Points Discussed during the Rebuttal**.

### **a. Reviewers’ Overall Review:**

According to extensive community feedback, the severe OpenReview bug occurred between **28 Nov, 23:00 and 00:00 (29 Nov)**. **The final discussion** message was posted on **28 Nov at 04:57**, which is approximately **18 hours before the bug appeared**. Therefore, **all rebuttal exchanges between us and the reviewers took place entirely before the incident occurred**.

Below, we summarize each reviewer's score and participation along with the timestamps:

* **Reviewer rfAK**: Initial score **4**; did not participate in the discussion.
* **Reviewer 9ULh**: Initial score **6**; did not participate in the discussion.
* **Reviewer 8395**: Initial score **6**; participated in the discussion and maintained the same score.

> 8395 (26 Nov, 12:39)：Thanks for the clear replies. I read through your explanations and also checked the discussions with the other reviewers. I think you addressed the main points well. I’ll keep my original score (6, borderline accept).

**Reviewer VvkZ:** Initial score 2. The reviewer actively participated in the discussion and subsequently **raised the score to 6**.

> VvkZ (27 Nov, 05:17): The authors have provided a comprehensive response to Q1 with experiments that are convincing. There response to Q2 is also convincing. I do not find their responses to Q3 and Q4 completely persuasive, but I do not think that these issues are as critical. I have raised my score to 6.

In summary, our final scores are **4-6-6-6**. We would like to emphasize that although we have thoroughly addressed all concerns raised by **Reviewer rfAK (score 4)**, the reviewer **did not engage in the discussion** throughout the process. As a result, the initial score remains unsubstantiated by subsequent dialogue, **leaving the initial score of 4 unsupported**.

---

### **b. Key Points Discussed during the Rebuttal:**

Below we summarize the key points raised by the four reviewers during the discussion phase, along with our corresponding resolutions.

### **1. Discussion of existing content:**

**1)** Clarifying the importance of visual prompts — how often do users actually ask about people or objects that are difficult to describe? (reviewer VvkZ)

We updated the interaction workflow and conducted a new experiment to examine it. Results show that **users naturally rely on visual prompts**. The reviewer found this experiment **convincing**. See **Table 3** and **Appendix E.2** for details.

**2)** Dataset bias — do the questions in the dataset favor visual prompts? (reviewer VvkZ)

We provided a detailed explanation of the dataset construction process, demonstrating that the dataset does not favor visual prompts. Details are discussed in **Appendix A.6**.

### **2. Scalability Discussion:**

**1)** The impact of open-ended QA on hack behaviors (reviewer 9ULh)

We first note that existing benchmarks use MCQ because it is **easy to evaluate, highly controllable, and avoids ambiguity in open-ended responses**. Second, two experiments show that **open-ended generation largely mitigates hack behaviors**. Details are provided in **Appendix C.2**.

**2)** How does the structure of the visual query itself affect performance? (reviewer 9ULh)

We conducted experiments to investigate the impact of visual query structure on the effectiveness of visual prompts. Detailed results can be found in **Section 4.2**.

**3)** Using temporal certificates or frame-rate ablations to quantify the impact of frame count on performance (reviewer 9ULh, VvkZ)

Following EgoSchema, we compute the **temporal certificate length** and the **temporal certificate ratio** based on the video length and task type in our dataset. We also evaluate model performance across task types and video durations. Further details are provided in **Appendix A.5 and C.3**.

### **3. Paper already covered but possibly overlooked by reviewers:**

**1)** Lack of evidence that visual prompts align better with human cognition (reviewer 8395)

In **Section 4.2**, we designed two experiments and results show that **visual prompts are both more model-friendly and more user-friendly than text prompts**.

**2)** Providing quantitative evaluation schemes for user experience assessment and hack-behavior detection (reviewer rfAK)

Our paper already includes a **user experience evaluation** (*P7 Table 3*) and **hack-behavior detection experiments** (*P8 Figure 5, P24 Figure 13*).

We hope that this summary helps facilitate your understanding of our work and supports your decision-making process. Thank you again for your valuable time and effort.

Sincerely,
Authors of Paper 8969

---

### Meta-Review · Area_Chair_cEWY · 2025-12-25

**Summary:**

Reviewer rfAK noted that the benchmark focuses on synthetic QA tasks rather than natural, conversational dialogues, which limits its application to real-world scenarios, and that the benchmark evaluates static performance but does not assess how models adapt or progress over time.

Reviewers 8395 and 9ULh raised the lack of deeper investigations into why models fail at spatiotemporal reasoning and how the benchmark effectively eliminates or identifies hack behaviors.

**Reviewer Concerns:**

The authors conducted a new user study with 20 volunteers, showing that 64.5% preferred visual prompts and 68% of questions were naturally initiated with them. This addressed Reviewer VvkZ and 8395's requests for empirical evidence of human alignment.

The authors converted the benchmark to an open-ended (OE) format and added experiments to address the concerns on hack behaviors.

The authors quantified temporal certificates to show deeper analysis on spatiotemporal reasoning. They showed that the Basic Perception (BP) tasks can be solved with one frame. Temporal Understanding (TU) and High-level Reasoning (HR) tasks show significant performance gains as the sampling frame rate increases.

Reviewer rfAK noted that the benchmark focuses on synthetic QA rather than natural, free-form multimodal dialogues. While the authors showed consistency between MCQ and open-ended results, the benchmark remains a static evaluation tool rather than a truly interactive dialogue system.

Reviewers 8395 and VvkZ highlighted the lack of audio or other sensory signals. The authors acknowledged this as an inherent limitation, deferring the inclusion of full-modality inputs to a future version.

**Reviewer Scores:**

Reviewer rfAK's main points on weaknesses are scalability and the lack of quantitative frameworks. The authors addressed rfAK's concerns on quantitative metrics by highlighting Table 3 (User Experience) and Figure 5 (Hack Behavior). They also proposed an automated pipeline to address the scalability concern. Given these rebuttal, this reviewer is likely to increase the final rating to 6, or maintain the rating at 4 with a lower chance.

Other 3 reviewers expressed their preference during the discussion/initial reviews, and maitained/changed the ratings to 6.

---

### Decision · Program_Chairs · 2026-01-26

Accept (Poster)